# BEYOND PAIRWISE: EMPOWERING LLM ALIGNMENT WITH (RANKED) CHOICE MODELING

**Yuxuan Tang**
Institute of Operations Research and Analytics
National University of Singapore
yuxuan.tang@u.nus.edu

**Yifan Feng**
Department of Analytics and Operations
NUS Business School
yifan.feng@nus.edu.sg

## ABSTRACT

Alignment of large language models (LLMs) has predominantly relied on pairwise preference optimization, where annotators select the better of two responses to a prompt. While simple, this approach overlooks the opportunity to learn from richer forms of human feedback, such as multiway comparisons and top-$k$ rankings. We introduce *Ranked Choice Preference Optimization* (RCPO), a unified framework that bridges preference optimization with (ranked) choice modeling via maximum likelihood estimation. RCPO supports both utility-based and rank-based models, subsumes several pairwise methods (such as DPO and SimPO) as special cases, and provides principled training objectives for richer feedback formats. We instantiate this framework with two representative models (Multinomial Logit and Mallows-RMJ). Experiments on Llama-3-8B-Instruct, Gemma-2-9B-it, and Mistral-7B-Instruct across in-distribution and out-of-distribution settings show that RCPO consistently outperforms competitive baselines. RCPO shows that directly leveraging ranked preference data, combined with the right choice models, yields more effective alignment. It offers an extensible foundation for incorporating (ranked) choice modeling into LLM training.

## 1 INTRODUCTION

Large language models (LLMs), a prominent form of generative AI, have rapidly transformed human-computer interaction, powering applications from open-ended dialogue (Thoppilan et al., 2022) and content creation (Brown et al., 2020) to code generation (Chen et al., 2021; Li et al., 2023) and healthcare decision support (Nori et al., 2023). A key driver of this success is *alignment*: training models to produce outputs that are factual, helpful, safe, and aligned with social norms.

Reinforcement learning from human feedback (RLHF) (Ziegler et al., 2019; Stiennon et al., 2020; Ouyang et al., 2022) has emerged as the dominant paradigm for aligning AI systems, exemplified by ChatGPT and GPT-4 (Achiam et al., 2023). More recently, Direct Preference Optimization (DPO) Rafailov et al. (2023) has achieved comparable results through a simpler and more efficient objective, and has been used to fine-tune LLMs such as Llama-3 (Dubey et al., 2024) and Zephyr (Tunstall et al., 2023). The success of RLHF and DPO has spurred a surge of research, resulting in numerous extensions (Zhao et al., 2024; Winata et al., 2025).

Despite their differences, most alignment methods rely on *pairwise preference data*, where for each prompt $x$, a preferred response $y_w$ and a dispreferred response $y_l$ are selected by human annotators or AI judges. In practice, however, preference feedback is often richer than simple pairs. Annotators may provide partial rankings, top-$k$ selections, or single-best judgments from a larger candidate set. Current approaches typically reduce this richer information to pairs—e.g., in training InstructGPT, Ouyang et al. (2022) collected rankings of $K$ responses per prompt but converted them into all $\binom{K}{2}$ pairs; in many academic alignment studies (Meng et al., 2024; Chen et al., 2025; Zhao et al., 2024; Gupta et al., 2025), multiple responses are scored by a reward model, but only the highest- and lowest-scoring are kept. These reductions, while convenient for pairwise-based algorithms, risk distorting the original preference structure and discarding potentially valuable information.

Figure 1: Ranked Choice Preference Optimization (RCPO)

To address this gap, we propose **R**anked **C**hoice **P**reference **O**ptimization (**RCPO**), a general framework that generalizes pairwise preference alignment to ranked choice feedback (see Figure 1). Rather than restricting evaluators to comparing only two responses, RCPO presents a set of candidates and asks them to choose either the single-best or the top-$k$ responses for a given prompt. Our approach is grounded in the theory of *(ranked) choice models*, particularly discrete choice modeling, which have been extensively studied in psychology, marketing, economics, and operations research. To the best of our knowledge, RCPO is among the first attempts to systematically apply (ranked) choice modeling to LLM alignment.

The main contributions of this paper are threefold:

(1) **Conceptual Framework**: We establish a systematic connection between LLM fine-tuning and choice modeling, showing that fine-tuning can be essentially reduced to maximum likelihood estimation (MLE) of choice models. Building on this insight, we develop RCPO as a principled extension of pairwise preference alignment that directly incorporates ranked choice feedback. RCPO is a general and flexible framework: any choice model that satisfies certain regularity conditions can be integrated. By preserving the richness of the original annotations – whether single-best or top-$k$ preferences – RCPO mitigates the information inefficiency in pairwise conversion and enables more effective alignment with human intent.

(2) **Concrete Examples**: We showcase how two broad classes of choice models, i.e., utility- or rank-based, can be accommodated within the RCPO framework. We then instantiate RCPO with a representative model from each class: the Multinomial Logit (MNL) model (McFadden, 1972) for utility-based choices and the Mallows-RMJ model (Feng & Tang, 2022) for rank-based choices. For both models, we derive alignment objectives under single-best and top-$k$ settings (see Table 1). We also use gradient analysis to shed light on the theoretical underpinnings of these preference optimization methods.

Table 1: Various Preference Optimization Objectives in RCPO

| Choice Model | Preference | Objective |
|---|---|---|
| MNL | Pairwise (DPO) | $-\log \sigma\left(f_\theta(x, y_w, y_l)\right)$ |
| | Single-Best | $-\log \sigma\left(-\log \sum_{y_i \in S \setminus \{y_w\}} \exp\left(f_\theta(x, y_i, y_w)\right)\right)$ |
| | Top-$k$ Choice | $-\sum_{i=1}^{k} \log \sigma\left(-\log \sum_{y_j \in S \setminus \{y_1, \ldots, y_i\}} \exp\left(f_\theta(x, y_j, y_i)\right)\right)$ |
| Mallows-RMJ | Pairwise | $-\log \phi(x) \cdot \sigma\left(f_\theta(x, y_l, y_w)\right)$ |
| | Single-Best | $-\log \phi(x) \cdot \sum_{y_i \in S \setminus \{y_w\}} \sigma\left(f_\theta(x, y_i, y_w)\right)$ |
| | Top-$k$ Choice | $-\log \phi(x)\left(\sum_{i=1}^{k-1}(|S| - i)\,\sigma\left(f_\theta(x, y_{i+1}, y_i)\right) + \sum_{y_j \in S \setminus \{y_1, \ldots, y_k\}} \sigma\left(f_\theta(x, y_j, y_k)\right)\right)$ |

**Notes:** $f_\theta(x, y_1, y_2) := \beta \log \frac{\pi_\theta(y_1 | x)}{\pi_{\text{ref}}(y_1 | x)} - \beta \log \frac{\pi_\theta(y_2 | x)}{\pi_{\text{ref}}(y_2 | x)}$.

(3) **Experiments**: We evaluate RCPO on state-of-the-art instruction-tuned LLMs (Llama-3-8B-Instruct, Gemma-2-9B-it, and Mistral-7B-Instruct) using in-distribution test and widely adopted out-of-distribution benchmarks (AlpacaEval 2 and Arena-Hard). Results consistently show that RCPO improves model performance and demonstrates flexibility across base models, preference feedback, and evaluation settings. We highlight the Mallows-RMJ-based preference optimization, which achieves strong results under both pairwise and ranked choice setups.

Collectively, these contributions position RCPO as a principled and practical framework for leveraging ranked choice feedback to advance the alignment of large language models.

## 2 A CHOICE BASED ALIGNMENT FRAMEWORK

### 2.1 RLHF AND DPO

We start by recapping the key concepts in RLHF and DPO. Let $x$ denote an input prompt and $y$ denote a candidate response. A language model is parameterized by a policy $\pi_\theta$, where $\pi_\theta(y \mid x)$ represents the probability of generating response $y$ given prompt $x$.

RLHF comprises three sequential phases. First, it fine-tunes a pre-trained LLM through supervised learning on supervised data and produces an SFT model, which we use as the reference policy $\pi_{\text{ref}}$. Second, RLHF fits a reward model $r^*(x, y)$, which can be a neural network itself, based on a separate pairwise preference dataset $\mathcal{D} = \{(x^{(i)}, y_w^{(i)}, y_l^{(i)})\}_{i=1}^N$. Here $x^{(i)}$ represents the prompt provided to annotator $i$, and $y_w^{(i)}$ and $y_l^{(i)}$ are the preferred and dispreferred responses, respectively. Third, the LLM is further fine-tuned via reinforcement learning to maximize the regularized expected reward:

$$\max_{\pi_\theta} \mathbb{E}_{x\sim\mathcal{D}}\big[\mathbb{E}_{y\sim\pi_\theta(y|x)}\big[r^*(x,y)\big] - \beta\,\mathrm{KL}\big(\pi_\theta(\cdot \mid x)\,\|\,\pi_{\text{ref}}(\cdot \mid x)\big)\big], \tag{1}$$

where $\beta > 0$ is a hyperparameter controlling the deviation from the reference policy $\pi_{\text{ref}}$.

While RLHF has shown impressive results, both reward model fitting and reinforcement learning require substantial computational effort. In this light, DPO analytically solves (1), yielding a closed-form relationship between the optimal policy and the reward model given by:

$$r^*(x,y) = \beta \log \tfrac{\pi_{\theta*}(y|x)}{\pi_{\text{ref}}(y|x)} + \beta \log Z(x), \tag{2}$$

where $Z(x) = \sum_y \pi_{\text{ref}}(y \mid x) \exp(r^*(x,y)/\beta)$ is the partition function. As a result, based on a Bradley-Terry preference assumption on how the preferred/dispreferred responses are generated, DPO consolidates the last two steps of RLHF into a direct optimization problem with the following loss function:

$$\min_{\pi_\theta} \mathcal{L}_{\text{DPO}}\big(\pi_\theta; \pi_{\text{ref}}\big) = -\mathbb{E}_{(x,y_w,y_l)\sim\mathcal{D}}\big[\log\sigma\big(\beta\log\tfrac{\pi_\theta(y_w|x)}{\pi_{\text{ref}}(y_w|x)} - \beta\log\tfrac{\pi_\theta(y_l|x)}{\pi_{\text{ref}}(y_l|x)}\big)\big], \tag{3}$$

where $\sigma(\cdot)$ is the sigmoid function. In other words, DPO directly fine-tunes $\pi_\theta$ to match human pairwise preferences in a single step, thereby greatly reducing computational overhead.

### 2.2 (RANKED) CHOICE MODELING

**Discrete Choice Models.** We also introduce the key concepts for (ranked) choice modeling. First, a rich body of research across economics, marketing, and operations research has developed a variety of discrete choice models to represent human preferences. Let $\mathcal{N} = \{y_1, y_2, \ldots, y_n\}$ denote the universe of items. A *discrete choice model* specifies the probability of selecting an *item* $y$ from an *assortment* $S \subseteq \mathcal{N}$ under a given *context* $x$, denoted by $\mathbb{P}(y \mid S; x)$. This is an abstraction of many business and economics use cases. For instance, in retail settings, items typically correspond to products, assortments represent the sets of available products at the time of choice, and the context encompasses covariates such as prices, promotions, or product features. There are many ways to define a discrete choice model, which essentially reduce to specifying the probability function $\mathbb{P}(y \mid S; x)$.

**Ranked Choice Models.** Discrete choice models can be extended to richer feedback in the form of *ranked choices*. We adopt the ranked choice framing of Feng & Tang (2022). Let $\mu^k = y_1 \succ y_2 \succ \cdots \succ y_k$ with $\{y_1, \ldots, y_k\} \subseteq S$ denote a top-$k$ list of items from $S$. A ranked choice model defines $\mathbb{P}(\mu^k \mid S; x)$, which specifies the probability of observing the partial ranking $\mu^k$ under context $x$. This notion of ranked choices generalizes several common feedback structures, such as (i) pairwise comparisons; (ii) discrete choices (or multiway comparisons); (iii) listwise feedback (or full rankings over the items in any given assortment); (iv) top-$k$ rankings over a full item set, to name a few. For example, when $k = 1$, a ranked choice model reduces to a discrete choice model.

**Assumptions.** In this paper, we will put (ranked) choice models in the context of LLM alignment, and focus on those that satisfy the following two assumptions:

- *Assumption A1: Reward sufficiency.* There exists a real-valued reward function $r : \mathcal{X} \times \mathcal{Y} \to \mathbb{R}$ and a mapping $g$ such that, for every $S$ and $x$,

$$\mathbb{P}(\mu^k \mid S \,; x) \;=\; g\big(\mu^k, S, \{\, r(x,y) \,\}_{y \in S}\big), \text{ for every } \mu^k = y_1 \succ y_2 \succ \cdots \succ y_k \text{ within } S.$$

That is, the effect of $x$ and $S$ on ranked choice probabilities enters *only* through the item rewards $\{r(x,y)\}_{y \in S}$. Note that in the special case of discrete choices, the condition above can be simplified to $\mathbb{P}(y \mid S \,; x) = g(y, S, \{\, r(x,y') \,\}_{y' \in S})$, for every $y \in S$.

- *Assumption A2: MLE estimability.* Given observations $\mathcal{D} = \{(x_i, S_i, \mu_i^k)\}_{i=1}^N$, the rewards are identifiable up to the usual invariances (e.g., additive shift and positive scaling) and admit tractable log-likelihood function

$$\sum\nolimits_{i=1}^N \log g\big(\mu_i^k, S_i, \{\, r(x_i, y) \,\}_{y \in S_i}\big).$$

Here the meaning of "tractable" will be clearer later. In short, the log-likelihood function should admit simple gradients to be passed to the training of $\pi_\theta$.

As can be easily seen, many choice models satisfy the assumptions above. In Sections 2.4 and 2.5, we present two approaches: utility-based models and rank-based models.

## 2.3 RCPO: Connection between LLM Alignment and Ranked Choice Modeling

Our framework starts with a conceptual insight into LLM alignment: if we interpret a prompt $x$ as the *context*, a candidate response $y$ as an *item*, and a set of candidate responses as an *assortment* $S$, then every choice model offers a distinct way to incorporate annotators' preference feedback via an MLE objective. For instance, consider the case where $S = \{y_w, y_l\}$, and let $g$ denote the Bradley-Terry choice rule. Then the probability that $y_w$ is preferred over $y_l$ is given by:

$$\mathbb{P}(y_w \mid \{y_w, y_l\} \,; x) = g(y_w, \{y_w, y_l\}, \{r(x,y_w), r(x,y_l)\}) = \sigma\big(r(x,y_w) - r(x,y_l)\big),$$

where $\sigma$ denotes the sigmoid function. Substituting the reward function defined in (2) into the Bradley–Terry likelihood yields the DPO objective described in (3). This establishes that DPO is a special case of our formulation, where preferences follow the Bradley-Terry pairwise comparison model.

**The RCPO Framework.** Motivated by these connections, we introduce a general framework for preference optimization grounded in choice model theory. Specifically, we extend DPO from pairwise Bradley-Terry comparisons to *arbitrary* ranked choice models that satisfy the assumptions outlined above. In this framework, once the functional forms of $r(x,y)$ are specified, and the choice rule $g$ is determined by a ranked choice model, the corresponding preference optimization procedure is defined by the following maximum likelihood estimation (MLE) objective:

$$\max_{\pi_\theta} \sum\nolimits_{i=1}^N \log g\left(\mu_i^k, S_i, \{r_{\pi_\theta}(x_i,y)\}_{y \in S_i}\right),$$

where $r_{\pi_\theta}(x,y)$ is a reward function derived from the policy $\pi_\theta$.

Beyond the reward function (2) used in DPO, the literature has proposed many alternative definitions. For example, Wang et al. (2023) introduces $f$-divergence generalizations, while Meng et al. (2024) and Gupta et al. (2025) propose length-normalized log-likelihoods. These alternative reward formulations can likewise be incorporated into the RCPO framework. Consequently, methods such as R-DPO (Park et al., 2024), SimPO (Meng et al., 2024), and AlphaPO (Gupta et al., 2025) can also be viewed as special cases of RCPO. In particular, while these approaches adopt the Bradley–Terry choice rule, they differ in the functional form of the reward $r(x,y)$. Although our framework can accommodate a wide range of reward functions, for clarity, we restrict attention to the reward defined in (2) thereafter in the paper.

What *truly* demonstrates the versatility of RCPO is how it gives birth to new preference optimization methods. In this paper, we consider two prominent classes of choice models: utility-based models, which rely on the numerical magnitudes of the items' utilities, and rank-based models, which depend on the rankings of the items. Despite using different representations of the underlying preference distribution, they are both compatible with the unified RCPO framework.

## 2.4 UTILITY-BASED CHOICE MODELS

The **random utility model (RUM)** is perhaps the most widely studied class of discrete choice models. Originally proposed by Thurstone (2017), it has been extensively developed in the economics and operations management literature (Anderson et al., 1992; Train, 2009). RUM assumes that every item $y_i \in \mathcal{N}$ comes with a *utility*, which takes the form $u_{y_i} = \nu_{y_i} + \varepsilon_{y_i}$, where $\nu_{y_i}$ is the mean utility, and $\varepsilon_{y_i}$ is an exogenous random utility shock term. In this paper, we also write $u_{y_i}$ and $\nu_{y_i}$ in the form of $u_{y_i}(x)$ and $\nu_{y_i}(x)$ to emphasize that they can be context-dependent. Given the mean utility vector $\nu(x) = (\nu_{y_1}(x), \dots, \nu_{y_n}(x))$ and a distribution $f$ over the utility shocks $\varepsilon = (\varepsilon_{y_1}, \dots, \varepsilon_{y_n})$, the probability of choosing alternative $y_i$ from an assortment $S \subseteq \mathcal{N}$ is

$$\mathbb{P}(y_i \mid S\,;x) = \int_{\varepsilon} \mathbb{I}\{\nu_{y_i}(x) - \nu_{y_j}(x) > \varepsilon_{y_j} - \varepsilon_{y_i} \ \ \forall y_j \in S \setminus \{y_i\}\} f(\varepsilon)\,\mathrm{d}\varepsilon. \tag{4}$$

RUMs are categorized by the distribution of their stochastic terms. The multinomial logit (MNL) model, introduced by McFadden (1972), assumes i.i.d. Gumbel noise. Alternatives include the probit model (joint normal distribution (Daganzo, 2014)), the nested logit model (correlated extreme value distributions (McFadden, 1980)), and the exponomial model (negative exponential distributions (Alptekinoğlu & Semple, 2016)).

The RUM can be extended to a ranked choice model, where the probability of observing the top-$k$ ranking $\mu^k = y_1 \succ y_2 \succ \cdots \succ y_k$ from an assortment $S$ is

$$\mathbb{P}(\mu^k \mid S\,;x) = \int_{\varepsilon} \prod_{\ell=1}^{k} \mathbb{I}\Big\{\nu_{y_\ell}(x) - \nu_{y_j}(x) > \varepsilon_{y_j} - \varepsilon_{y_\ell},\ \forall y_j \in S \setminus \{y_1, \dots, y_\ell\}\Big\} f(\varepsilon)\,\mathrm{d}\varepsilon. \tag{5}$$

As (4) and (5) show, the choice probabilities depend only on the mean utility vector $\nu(x)$, which corresponds to the reward vector in our framework. Thus, any RUM that admits MLE estimation of $\nu(x)$ can be incorporated into LLM fine-tuning.

## 2.5 RANK-BASED CHOICE MODELS

While utility-based models operate on numerical utilities, rank-based models represent preferences as complete orderings over $\mathcal{N}$, depending only on the relative positions of items[1]. See Jagabathula & Venkataraman (2022) for a survey of such models.

The **Mallows-type model** (Mallows, 1957; Fligner & Verducci, 1986) is among the most widely used classes of ranking models. Let $\mathfrak{S}_n$ be the set of permutations of $\mathcal{N}$. A Mallows-type model assigns a probability distribution over permutations $\mu \in \mathfrak{S}_n$ based on their distance from a central ranking $\mu_0$:

$$\mathbb{P}_{\phi,\mu_0,d}(\mu) := \frac{\phi^{d(\mu_0,\mu)}}{\sum_{\mu'} \phi^{d(\mu_0,\mu')}}, \quad \mu \in \mathfrak{S}_n,$$

where $d(\cdot,\cdot)$ is a distance function between permutations, $\phi \in (0,1)$ is a dispersion parameter. Intuitively, rankings closer to $\mu_0$ are exponentially more likely. Different distance choices yield different variants, such as Kendall's Tau (Mallows, 1957), Spearman's rank and footrule (Diaconis & Graham, 1977), Hamming distance (Bookstein et al., 2002), Cayley distance (Irurozki et al., 2018), and Reverse Major Index (Feng & Tang, 2022). To capture context dependence, we parameterize the central ranking and dispersion as functions of $x$, denoted $\mu_0(\cdot|x)$ and $\phi(x)$. For readability, we may suppress $x$ in the notation when its dependence is unambiguous.

For a ranking $\mu \in \mathfrak{S}_n$ and an item $y$, denote $\mu^{-1}(y)$ as the position of $y$ in $\mu$ (smaller means higher preference). Given an assortment $S \subseteq \mathcal{N}$ and a top-$k$ ranking $\mu^k = y_1 \succ y_2 \succ \cdots \succ y_k$ with $\{y_1, \dots, y_k\} \subseteq S$, the implied ranked choice probability of observing $\mu^k$ from $S$ is given by

$$\mathbb{P}(\mu^k \mid S\,;x) = \sum_{\mu \in \mathfrak{S}_n} \mathbb{P}_{\phi(x),\mu_0(\cdot|x),d}(\mu)\, \mathbb{I}\{\mu, \mu^k, S\}, \tag{6}$$

where $\mathbb{I}\{\mu, \mu^k, S\} = 1$ if $\mu$ ranks $\{y_1, \dots, y_k\}$ in the specified order, and each of them is ranked above all remaining items in $S$, or more formally, $\mathbb{I}\{\mu, \mu^k, S\} := \mathbf{1}\left\{ \begin{matrix} \mu^{-1}(y_1) < \cdots < \mu^{-1}(y_k), \\ \mu^{-1}(y_\ell) < \mu^{-1}(y_j)\ \forall y_j \in S \setminus \{y_1, \dots, y_k\},\ \forall \ell \in \{1, \dots, k\} \end{matrix} \right\}$. When $k = 1$, this reduces to a standard discrete choice probability. If $\phi(x)$ is known, the choice probability depends only on $\mu_0(\cdot|x)$, which can be represented as a vector of normalized ranks. Consequently, any Mallows-type model that supports MLE estimation of $\mu_0(\cdot|x)$ can be embedded within our framework.

---

[1]Although technically, an RUM can be transformed into a distribution of rankings, not every RUM can necessarily be transformed into a Mallows-type model. Therefore, we treat them separately in this paper, with different ways to parametrize the choice probabilities.

## 3    TWO EXAMPLES OF THE RCPO FRAMEWORK

In this section, we consider two representative examples: the Multinomial Logit (MNL) model as a utility-based choice model, and the Mallows-RMJ model as a rank-based choice model. These models are selected for the simplicity of their choice probabilities and the tractability of MLE. For each model, we derive the corresponding objectives for both single-best and top-$k$ feedback, demonstrating the framework's flexibility across diverse preference structures.

### 3.1    MULTINOMIAL LOGIT MODEL (MNL)

**Discrete Choice.**    If we take the random shock $\varepsilon$ to be i.i.d. Gumbel, we recover the Multinomial Logit (MNL) model (McFadden, 1972), arguably the most widely used RUM. The corresponding choice probabilities in (4) admit simple closed-form expressions, given by $\mathbb{P}(y_i|S; x) = e^{\nu_{y_i}(x)} / \sum_{j=1}^{|S|} e^{\nu_{y_j}(x)}$. As such, it naturally extends the Bradley-Terry model from pairwise comparisons to multi-item choice settings. By representing the mean utility as the reward defined in (2), we have the following theorem.

**Theorem 1 (MNL-PO-Discrete)** *Suppose the underlying single-best choice preference distribution follows MNL, the corresponding policy optimization objective is given by:*

$$\min_{\pi_\theta} \; -\mathbb{E}_{(x,S,y_w)\sim\mathcal{D}} \log \sigma\Big(-\log \sum_{y_i \in S\setminus\{y_w\}} \exp\big(\beta \log \tfrac{\pi_\theta(y_i|x)}{\pi_{\text{ref}}(y_i|x)} - \beta \log \tfrac{\pi_\theta(y_w|x)}{\pi_{\text{ref}}(y_w|x)}\big)\Big). \tag{7}$$

As in vanilla DPO, $\beta$ controls the deviation from the reference policy. When empirically solving (7), the size of each prompt's assortment $S$ is allowed to vary, enabling the sampling of additional responses, and thus finer-grained preferences, for certain prompts to further enhance alignment. Similar objectives appear in Ziegler et al. (2019) and Chen et al. (2024), where they are treated as softmax loss functions rather than being interpreted through the lens of discrete choice model.

**Top-$k$ Ranked Choice.**    As shown in Feng & Tang (2023), the MNL model also extends to a simple ranked choice model. Specifically, the probability that a top-$k$ ranked choice $\mu^k = y_1 \succ y_2 \succ \ldots \succ y_k$ is chosen out of an assortment $S \subseteq \mathcal{N}$ is given by

$$\mathbb{P}(\mu^k \mid S\,;x) = \prod_{i=1}^{k} \frac{e^{\nu_{y_i}(x)}}{\sum_{j=i}^{k} e^{\nu_{y_j}(x)} + \sum_{y_h \in S\setminus\{y_1,\ldots,y_k\}} e^{\nu_{y_h}(x)}}. \tag{8}$$

Hence we can write down the corresponding policy optimization objective as follows.

**Theorem 2 (MNL-PO-Top-k)** *Suppose the underlying top-$k$ choice preference distribution follows (8), the corresponding policy optimization objective is given by:*

$$\min_{\pi_\theta} \; -\mathbb{E}_{(x,S,\mu^k)\sim\mathcal{D}} \sum_{i=1}^{k} \log \sigma\Big(-\log \sum_{y_j \in S\setminus\{y_1,\ldots,y_i\}} \exp\Big(\beta \log \tfrac{\pi_\theta(y_j|x)}{\pi_{\text{ref}}(y_j|x)} - \beta \log \tfrac{\pi_\theta(y_i|x)}{\pi_{\text{ref}}(y_i|x)}\Big)\Big).$$

### 3.2    MALLOWS-RMJ MODEL

**Discrete Choice.**    A notable challenge in applying Mallows-type models to (ranked) choice modeling is that the ranked choice probabilities in (6) are usually difficult to obtain, since the sum is taken over all permutations. In this regard, an exception is Feng & Tang (2022), who adopt a Mallows-type model using the Reverse Major Index (RMJ) as the distance function. They show that, unlike other Mallows-type models, the Mallows-RMJ distribution admits a closed-form expression for the choice probabilities derived from (6):

$$\mathbb{P}(y_i \mid S\,;x) = \frac{\phi(x)^{d(y_i,S)}}{1+\phi(x)+\cdots+\phi(x)^{|S|-1}}, \tag{9}$$

where $d(y_i, S) := \sum_{y_j \in S\setminus\{y_i\}} \mathbb{I}\{\mu_0^{-1}(y_i \mid x) > \mu_0^{-1}(y_j \mid x)\}$ equals the number of items in $S$ that are ranked higher than $y_i$ according to the global ranking $\mu_0(\cdot \mid x)$. In other words, $d(y_i, S)$ defines the *relative ranking position* of item $y_i$ within the assortment $S$, so that its choice probability decays exponentially with its rank position in $S$.

A notable feature of this model is its exclusive dependence on *ordinal information*. For instance, when the assortment size is restricted to two, the model reduces to the classic *noisy comparison*

model, in which the superior item is chosen with a fixed probability $1/(1+\phi(x))$, independent of the absolute difference between the options. This is in contrast to the Multinomial Logit (MNL) model, where choice probabilities are directly tied to the cardinal utilities of items. This reliance on relative rankings, rather than precise utility estimates, provides the Mallows-RMJ model a form of *robustness* against model misspecification and noise in preference feedback. Such robustness may help explain the model's favorable empirical performance observed in our experiments (Section 4).

We next derive the corresponding policy optimization objectives. By normalizing the negative rank as the reward defined in (2), i.e., $-\mu_0^{-1}(y|x) = \beta \log \frac{\pi_\theta(y|x)}{\pi_{\text{ref}}(y|x)} + \beta \log Z(x)$, we have the following.

**Theorem 3 (Mallows-RMJ-PO-Discrete)** *Suppose the underlying single-best choice preference distribution follows (9), the corresponding policy optimization objective is given by:*

$$\min_{\pi_\theta} -\mathbb{E}_{(x,S,y_w)\sim\mathcal{D}}\Big[\log\phi(x)\cdot\sum_{y_i\in S\setminus\{y_w\}}\mathbb{I}\{\beta\log\tfrac{\pi_\theta(y_w|x)}{\pi_{\text{ref}}(y_w|x)} - \beta\log\tfrac{\pi_\theta(y_i|x)}{\pi_{\text{ref}}(y_i|x)} < 0\}\Big]. \tag{10}$$

We also derive the alignment objective for the pairwise comparison case, with details in Section C.

**Top-$k$ Ranked Choice.** The top-$k$ ranked choice probability under the Mallows-RMJ model is also given by a simple expression:

$$\mathbb{P}(\mu^k \mid S; x) = \tfrac{\psi(|S|-k,\phi(x))}{\psi(|S|,\phi(x))}\cdot\phi(x)^{d(\mu^k,S)}, \tag{11}$$

where $d(\mu^k,S) = \sum_{i=1}^{k-1}\mathbb{I}\{\mu_0^{-1}(y_i\mid x) > \mu_0^{-1}(y_{i+1}\mid x)\}(|S|-i) + \sum_{y_j\in S\setminus\{y_1,...,y_k\}}\mathbb{I}\{\mu_0^{-1}(y_k\mid x) > \mu_0^{-1}(y_j\mid x)\}$ and $\psi(n,\phi(x)) = \prod_{i=1}^n(1+\ldots+\phi(x)^{i-1})$. The corresponding policy optimization objective is thus given by the result below.

**Theorem 4 (Mallows-RMJ-PO-Top-k)** *Suppose the underlying top-$k$ choice preference distribution follows (11), the corresponding policy optimization objective is given by:*

$$\min_{\pi_\theta} -\mathbb{E}_{(x,S,\mu^k)\sim\mathcal{D}}\Bigg[\log\phi(x)\bigg(\sum_{i=1}^{k-1}(|S|-i)\,\mathbb{I}\Big\{\beta\log\tfrac{\pi_\theta(y_i|x)}{\pi_{\text{ref}}(y_i|x)} - \beta\log\tfrac{\pi_\theta(y_{i+1}|x)}{\pi_{\text{ref}}(y_{i+1}|x)} < 0\Big\}$$
$$+ \sum_{y_j\in S\setminus\{y_1,...,y_k\}}\mathbb{I}\Big\{\beta\log\tfrac{\pi_\theta(y_k|x)}{\pi_{\text{ref}}(y_k|x)} - \beta\log\tfrac{\pi_\theta(y_j|x)}{\pi_{\text{ref}}(y_j|x)} < 0\Big\}\bigg)\Bigg]. \tag{12}$$

To make Mallows-RMJ objectives practical for LLM training, we also overcome two challenges along the way: estimating the unknown dispersion parameter $\phi(x)$ and handling the step functions in the objectives that hinder optimization. For the first challenge, we follow a similar approach to Chen et al. (2025) and adapt the entropy proxy of $-\log\phi(x)$. For the second challenge, we smooth the step functions by replacing them with sigmoid approximations, which preserve the preference-based structure while yielding smoother, more informative gradients. More details are in Section D. The resulting objectives are summarized in Table 1.

### 3.3 GRADIENT ANALYSIS

To gain a deeper mechanistic understanding of the RCPO framework, we analyze the gradient structure of its loss functions. We focus here on the gradient of the Mallows-RMJ-PO-Top-k objective as a representative example. Full derivations and gradient expressions for other settings are deferred to Section F. The gradient with respect to model parameters $\theta$ is given by:

$$\nabla_\theta\mathcal{L}_{\text{Mallows-RMJ-PO-Top-}k}(\pi_\theta)$$

$$= \beta\,\mathbb{E}\Bigg[\underbrace{-\log\phi(x)}_{\substack{\text{greater weight for}\\\text{low-dispersion prompts}}}\bigg(\sum_{i=1}^{k-1}\underbrace{(|S|-i)}_{\substack{\text{greater weight for}\\\text{higher ranks}}}\underbrace{\sigma\big(f_\theta(x,y_{i+1},y_i)\big)\big(1-\sigma\big(f_\theta(x,y_{i+1},y_i)\big)\big)}_{\substack{\text{greater weight when}\\\text{rewards are similar}}}$$

$$\times\left(\underbrace{\nabla_\theta\log\pi_\theta(y_{i+1}\mid x)}_{\text{discourage }y_{i+1}} - \underbrace{\nabla_\theta\log\pi_\theta(y_i\mid x)}_{\text{encourage }y_i}\right)$$

$$+ \sum_{y_j\in S\setminus\{y_1,...,y_k\}}\underbrace{\sigma\big(f_\theta(x,y_j,y_k)\big)\big(1-\sigma\big(f_\theta(x,y_j,y_k)\big)\big)}_{\text{comparison difficulty}}\left(\underbrace{\nabla_\theta\log\pi_\theta(y_j\mid x)}_{\text{discourage }y_j} - \underbrace{\nabla_\theta\log\pi_\theta(y_k\mid x)}_{\text{encourage }y_k}\right)\bigg)\Bigg],$$

where the expectation is taken over ranked choice triples $(x, S, \mu^k) \sim \mathcal{D}$. Intuitively, this gradient update increases the likelihood of higher-ranked responses while decreasing that of lower-ranked ones. The magnitude of the update is amplified when: (i) the prompt context exhibits low dispersion (i.e., more confident preferences); (ii) the response occurs in a higher-ranked position; and (iii) the reward estimates are close, indicating a more informative comparison. As such, these properties drive the model to sharpen distinctions near the top of the ranking and better align its outputs with fine-grained preference structures.

## 4 EXPERIMENTS

In this section, we assess our methods on several widely used evaluation protocols. Additional experimental details are provided in Appendix G.

### 4.1 EXPERIMENTAL SETUP

We adopt Llama-3-8B-Instruct, Gemma-2-9B-it and Mistral-7B-Instruct as our fine-tuning bases, as they are widely used flagship instruction-tuned models that represent the state-of-the-art. We generate multiple responses to each prompt in the UltraFeedback train dataset (Cui et al., 2023) and use the Skywork-Reward-V2-Llama-3.1-8B reward model to provide feedback, which achieves state-of-the-art performance on seven major reward model benchmarks at the time of writing this paper (Liu et al., 2025a). We refer readers to Appendix G regarding details of constructing the ranking-based preference dataset.

**Evaluation.** We evaluate our fine-tuned models on out-of-distribution and in-distribution tests.

For the *out-of-distribution test*, we assess our models on two widely used instruction-following benchmarks: AlpacaEval 2.0 (Dubois et al., 2024) and Arena-Hard-v0.1 (Li et al., 2024). AlpacaEval 2.0 comprises 805 questions drawn from distinct datasets. Performance is measured by the win rate (WR) of model outputs against reference answers generated by GPT-4-Turbo. We also report the length-controlled win rate (LC), which adjusts WR to control for output length. Arena-Hard-v0.1 consists of 500 well-defined technical problem-solving prompts and evaluates models using WR against GPT-4-0314. Previous works (Meng et al., 2024) have demonstrated that Arena-Hard-v0.1 achieves stronger model separability than AlpacaEval 2.0. For both benchmarks, we use GPT-4.1-mini as the judge, replacing the default GPT-4-Turbo due to its improvements (OpenAI, 2025). To enhance cross-judge robustness, we additionally employ GPT-5-mini as the judge on Arena-Hard-v0.1. The results are in Section G.4.

For the *in-distribution test*, we first let each fine-tuned model generate responses on the UltraFeedback test dataset, and then compute its win rate against the preferred responses in the dataset, using GPT-4.1-mini as the judge. This methodology aligns with standard evaluation protocols in related literature (e.g., Rafailov et al., 2023; Liu et al., 2023; 2025b).

### 4.2 MAIN RESULTS

**Generalization performance.** All nine choice-based methods—four existing and five newly proposed RCPO variants— outperform other preference optimization baselines across most evaluation metrics. Notably, the best-performing RCPO method, Mallows-RMJ-PO-Top-2, surpasses the strongest non-RCPO baseline, IPO, by 4.00 percentage points on AlpacaEval LC, 19.5 percentage points on AlpacaEval WR, 6.2 percentage points on Arena-Hard WR, and 9.47 percentage points on UltraFeedback WR. These gains demonstrate the strong potential of choice modeling as a principled framework for aligning LLMs with human preferences.

**Impact of feedback structure.** Fixing the reward function as (2), we find that training on top-2 feedback generally leads to better performance than top-1. This reflects the benefit of richer feedback structure. We will discuss the impact of longer feedback structures in the later ablation study.

**Impact of choice models.** The selection of the choice model can have a substantial impact on performance. Fixing the reward function as (2), Mallows-RMJ performs strongly across feedback types. In the pairwise setting, it outperforms all baselines on AlpacaEval WR, Arena-Hard WR, and UltraFeedback WR, even surpassing MNL models trained on richer top-2 feedback. Its performance

Table 2: Evaluation Results for Llama-3-8B-Instruct.

| Method | AlpacaEval 2 | | Arena-Hard | UltraFeedback |
| --- | --- | --- | --- | --- |
| | LC (%) | WR (%) | WR (%) | WR (%) |
| Base Model | 24.76 (0.42) | 24.40 (1.44) | 23.6 (21.9,25.4) | 42.51 (1.04) |
| CPO (Xu et al., 2024) | 32.25 (0.31) | 34.18 (1.59) | 31.3 (29.5,33.0) | 58.71 (1.05) |
| IPO (Azar et al., 2024) | 37.95 (0.33) | 33.51 (1.63) | 31.0 (29.2,32.6) | 58.44 (1.05) |
| ORPO (Hong et al., 2024) | 35.01 (0.39) | 28.72 (1.52) | 27.5 (25.5,29.3) | 51.68 (1.06) |
| RRHF (Yuan et al., 2023) | 31.04 (0.21) | 25.36 (1.47) | 27.1 (25.3,28.9) | 44.28 (1.06) |
| SLiC-HF (Zhao et al., 2023) | 31.04 (0.21) | 25.36 (1.47) | 26.9 (25.2,28.9) | 44.09 (1.06) |
| KTO (Ethayarajh et al., 2024) | 32.55 (0.35) | 29.70 (1.54) | 25.8 (24.0,27.7) | 53.33 (1.06) |
| DPO (Rafailov et al., 2023) | 41.24 (0.35) | 40.24 (1.66) | 32.6 (30.6,34.7) | 62.36 (1.03) |
| R-DPO (Park et al., 2024) | 39.88 (0.34) | 38.01 (1.63) | 32.3 (30.1,34.3) | 60.68 (1.05) |
| SimPO (Meng et al., 2024) | **44.15** (0.25) | 38.84 (1.58) | 33.5 (31.3,35.8) | 50.17 (1.09) |
| DPO-AllPairs | 33.02 (0.24) | 38.47 (1.65) | 29.6 (27.3,31.5) | 51.95 (1.08) |
| Mallows-RMJ-PO-Pairwise | 39.33 (0.28) | 48.71 (1.67) | 36.5 (34.3,38.6) | 66.28 (1.02) |
| MNL-PO-Discrete | 41.33 (0.29) | 48.08 (1.68) | 35.6 (33.6,37.4) | 64.64 (1.03) |
| Mallows-RMJ-PO-Discrete | 39.19 (0.28) | 51.17 (1.67) | 36.3 (34.6,37.9) | 67.56 (1.01) |
| MNL-PO-Top-2 | 40.12 (0.22) | 47.69 (1.67) | 35.8 (33.2,38.2) | 64.01 (1.04) |
| Mallows-RMJ-PO-Top-2 | 41.95 (0.26) | **53.01** (1.68) | **37.2** (35.0,39.4) | **68.91** (1.00) |

**Notes:** Methods are grouped by horizontal rules. The first block reports the base model. The second block reports existing preference optimization baselines. The third block reports prior preference optimization methods that can be viewed as special cases within the RCPO framework. DPO-AllPairs denotes the DPO method trained on pairwise data implied from the Top-2 ranking. The final block reports new preference optimization variants proposed in this paper under the RCPO framework. Standard errors (in parentheses) are reported for AlpacaEval 2 and UltraFeedback, and 95% CIs (in parentheses) are reported for Arena-Hard WR.

further improves with discrete or top-2 choice data. We also observe that, when trained on the same feedback structure, the two choice models require nearly identical training time. Further details can be found in the Appendix G.3.

Taken together, our results highlight the promising potential of leveraging broader feedback structures, when paired with appropriate choice models, to achieve more effective alignment.

**Robustness check on Gemma-2-9B-it and Mistral-7B-Instruct.** As shown in Table 3, the Mallows-RMJ-PO-Top-2 consistently outperforms competitive baselines on the benchmarks, highlighting strong generalization across base models.

Table 3: Evaluation Results for Other Base Models.

| Method | Gemma-2-9B-it | | | Mistral-7B-Instruct | | |
| --- | --- | --- | --- | --- | --- | --- |
| | AlpacaEval 2 | | Arena-Hard | AlpacaEval 2 | | Arena-Hard |
| | LC (%) | WR (%) | WR (%) | LC (%) | WR (%) | WR (%) |
| Base Model | 46.74 (0.15) | 31.81 (1.58) | 43.3 (41.3,45.7) | 14.53 (0.44) | 11.94 (1.09) | 10.8 (9.5,12.0) |
| SimPO (Meng et al., 2024) | 54.11 (0.17) | 47.23 (1.68) | 57.4 (55.3,59.6) | 26.32 (0.38) | 30.13 (1.54) | **19.3** (17.8,20.6) |
| DPO (Rafailov et al., 2023) | **58.01** (0.18) | 56.13 (1.66) | 59.9 (57.6,62.2) | 23.32 (0.39) | 19.91 (1.34) | 16.8 (15.5,18.3) |
| Mallows-RMJ-PO-Top-2 | 55.64 (0.11) | **59.82** (1.65) | **60.9** (58.9,62.6) | **29.57** (0.22) | **37.58** (1.68) | 16.9 (15.4,18.3) |

**Notes:** Standard errors (in parentheses) follow AlpacaEval 2 metrics, and 95% CIs (in parentheses) follow Arena-Hard WR.

## 4.3 ABLATION STUDY

The ablation studies are conducted using the Llama-3-8B-Instruct. Due to space constraints, the main text focuses on the impact of the ranked choice length $k$ and the assortment size $|S|$. The sample efficiency of training on ranked choice data, the effects of the $\beta$ parameter, and different treatments on ranked responses are deferred to the Appendix G.5.

### 4.3.1 IMPACT OF RANKED CHOICE LENGTH $k$ AND ASSORTMENT SIZE $|S|$.

We study the impact of the ranked choice length $k$ and assortment size $|S|$ from two perspectives: generalization performance and time efficiency. The results are summarized in Table 4 below.

**Impact of $k$ on generalization performance.** A large $k$ may not necessarily improve performance. For example, when $|S| = 5$, $k = 2$ is the best. One possible reason is that higher values of $k$ make the alignment process more reliant on high-quality data. Specifically, when $k$ is small, selecting the top-$k$ items is relatively easy, but as $k$ becomes larger, distinguishing the relative order among the remaining candidates may become more difficult. Hence a too large $k$ can introduce additional noise or place unnecessary burden on annotators. This finding is conceptually consistent with the prior literature in a best-item selection context (Feng & Tang, 2023). It also highlights the benefit of top-$k$ feedback compared to the more extreme options, such as pairwise comparisons or full ranking.

**Impact of $|S|$ on generalization performance.** Increasing $|S|$ from two to three and five generally leads to better performance. Notably, even $|S| = 3$ can achieve significant improvement over pairwise ($|S| = 2$). One possible reason is that incorporating more negative samples enables the language model to learn better distinctions.

**Impact of $k$ and $|S|$ on time efficiency.** The training time is insensitive to the value of $k$. It scales approximately linearly with the assortment size $|S|$. The reason is that when $|S|$ is larger, fewer responses can be contained in one batch, and therefore more time to complete an epoch.

In general, we believe that a moderately small $(|S|, k)$ will stand in the sweet spot between keeping feedback simple and making efficient use of data.

Table 4: Evaluation Results of Varied $k$ and $|S|$.

| Method | AlpacaEval 2 LC (%) | AlpacaEval 2 WR (%) | Arena-Hard WR (%) | Training Time (Hours) |
|---|---|---|---|---|
| DPO | 41.24 (0.35) | 40.24 (1.66) | 32.6 (30.6, 34.7) | 1.5 |
| Mallows-RMJ-PO-Pairwise | 39.33 (0.28) | 48.71 (1.67) | 36.5 (34.3, 38.6) | 1.5 |
| assort3-Mallows-RMJ-PO-Discrete | 40.28 (0.27) | 50.49 (1.68) | 36.3 (34.1, 38.8) | 2.3 |
| assort3-Mallows-RMJ-PO-Top-2 | 39.57 (0.27) | 49.65 (1.68) | 36.9 (34.7, 39.0) | 2.3 |
| assort5-Mallows-RMJ-PO-Discrete | 39.19 (0.28) | 51.17 (1.67) | 36.3 (34.6, 37.9) | 3.5 |
| assort5-Mallows-RMJ-PO-Top-2 | **41.95 (0.26)** | **53.01 (1.68)** | 37.2 (35.0, 39.4) | 3.5 |
| assort5-Mallows-RMJ-PO-Top-3 | 41.05 (0.25) | 52.59 (1.69) | **37.7 (35.8, 39.8)** | 3.5 |
| assort5-Mallows-RMJ-PO-Top-4 | 39.92 (0.26) | 51.08 (1.68) | 37.4 (35.5, 39.2) | 3.5 |

**Notes:** Standard errors (in parentheses) follow AlpacaEval 2 metrics, and 95% confidence intervals (in parentheses) follow Arena-Hard WR.

## 5 CONCLUSION

In this work, we propose Ranked Choice Preference Optimization (RCPO), a general framework that connects preference optimization with choice model estimation. By leveraging maximum likelihood, RCPO unifies pairwise, single-best, and top-$k$ preference data within a principled formulation. Examples of both utility- and rank-based choice models, together with empirical evaluations, demonstrate that RCPO preserves richer feedback and improves alignment over pairwise-only methods. We hope this work provides a foundation for integrating more advanced choice models into LLM alignment and inspires future exploration of richer preference signals.

**Ethics statement**   This work adheres to the ICLR Code of Ethics. All authors have read and explicitly acknowledged compliance with the Code of Ethics during the submission process. Our study does not involve human subjects, personal or sensitive data, or any procedures that would raise ethical concerns. Therefore, we believe that this work poses no ethical risks.

**Reproducibility statement**   We have made significant efforts to ensure reproducibility. Complete proofs of all theoretical results and detailed derivations of the gradient analysis presented in Section 3 are provided in Section E and Section F, respectively. The full experimental details, including the computing environment, dataset construction process, the use of open-source models, hyperparameter settings, and evaluation protocols, are provided in the Section G. In addition, we provide the codes and training scripts in the supplementary materials. Together, these materials allow independent researchers to reproduce our results.

ACKNOWLEDGMENTS

This work is supported by the NUS Startup Grant (WBS Number: A-0003856-00-00), the Singapore Ministry of Education (MOE) Academic Research Fund (AcRF) Tier 1 (WBS Number: A-8003890-00-00), and the Singapore National Supercomputing Centre (NSCC) Young Investigator Seed Grant (Project ID: 31010027).

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

APPENDIX

## A   THE USE OF LARGE LANGUAGE MODELS (LLMS)

Large Language Models are used to assist in editing and polishing the writing. The authors take full responsibility for all content of the paper, and acknowledge that LLMs are not eligible for authorship.

## B   RELATED WORK

**Preference optimization.**   DPO's success has attracted significant research attention in the LLM alignment community, yielding numerous variants with alternative objectives. These include formulations based on ranking objectives (Yuan et al., 2023; Song et al., 2024; Liu et al., 2024; Zhao et al., 2025; Cai et al., 2025), pairwise comparisons employing other preference models (Chen et al., 2025), multi-level preference data (Zhang et al., 2024), other implicit reward formulations (Wang et al., 2023; Meng et al., 2024; Gupta et al., 2025), and approaches leveraging binary feedback on individual prompt-response pairs (Ethayarajh et al., 2024). Distinct from prior studies, we investigate novel forms of preference feedback that enable alignment methods to directly leverage richer choice-based signals.

**Discrete choice modeling.**   Research in marketing, economics, and operations research has studied various specifications of discrete choice models, including the multinomial logit model (McFadden, 1972), the general attraction model (Gallego et al., 2015), the Markov chain choice model (Blanchet et al., 2016), rank-based choice models (Farias et al., 2013), among others. These models play a critical role in informing key operational decisions such as inventory management, assortment planning, pricing, and matching optimization. For comprehensive overviews, we refer readers to Train (2009), Gallego et al. (2019), and Berbeglia et al. (2022). To the best of our knowledge, we are the first to apply discrete choice model theory in LLM alignment, enabling principled use of richer preference feedback beyond pairwise comparisons.

**Social choice and AI alignment.**   Social choice theory is a field of study that deals with the aggregation of individual preferences to form a collective decision. Current approaches to LLM alignment involves the experimental studies (Huang et al., 2024), conceptual frameworks (Prasad, 2018; Mishra, 2023; Dai & Fleisig, 2024; Conitzer et al., 2024; Zhi-Xuan et al., 2024), and axiomatic frameworks (Ge et al., 2024). In contrast to the standard social choice setting (Dai & Fleisig, 2024), where voters provide full rankings over all alternatives, we focus on aggregating ranked choice preferences.

## C   MALLOWS-RMJ IN PAIRWISE SETUP

**Pairwise Setup.**   When this model is restricted to the pairwise comparison setting, the preference probability simplifies to

$$\mathbb{P}\left(y_w \succ y_l \,;x\right) = \frac{\phi(x)^{\mathbb{I}\{\mu_0^{-1}(y_w|x) > \mu_0^{-1}(y_l|x)\}}}{1+\phi(x)}. \tag{13}$$

This formulation aligns with the widely studied noisy pairwise comparison models in the computer science literature, where only two items are compared at a time, and the preferred one is selected with a fixed probability that does not depend on the specific pair. The pairwise probability in (13) leads to our following optimization objective.

**Theorem 5 (Mallows-RMJ-PO-Pairwise)** *Suppose the underlying pairwise preference distribution follows (13), the corresponding policy optimization objective is given by:*

$$\min_{\pi_\theta} -\mathbb{E}_{(x,y_w,y_l)\sim\mathcal{D}}\big[\log\phi(x)\cdot\mathbb{I}\{\beta\log\tfrac{\pi_\theta(y_w|x)}{\pi_{\mathrm{ref}}(y_w|x)} - \beta\log\tfrac{\pi_\theta(y_l|x)}{\pi_{\mathrm{ref}}(y_l|x)} < 0\}\big]. \tag{14}$$

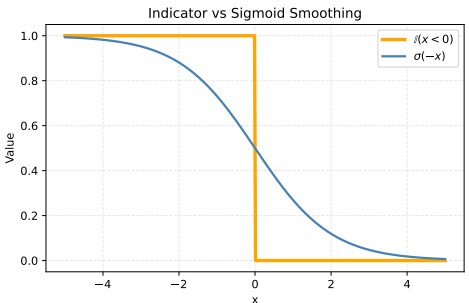
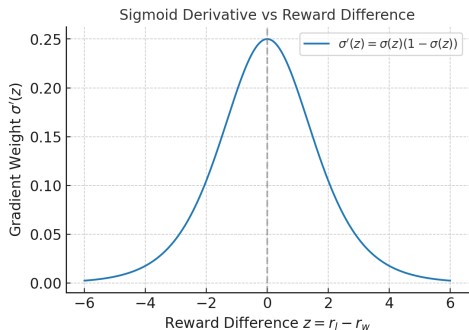

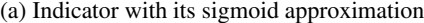

(a) Indicator with its sigmoid approximation.  (b) Derivative of sigmoid function.

## D  PRACTICAL SCHEME OF MALLOWS-RMJ METHODS

To make Mallows-RMJ-based objectives practical for LLM training, we address two key challenges: estimating the dispersion parameter $\phi(x)$ and stabilizing optimization via smooth approximations.

**A Token-level Entropy-based Proxy of** $-\log\phi(x)$ **for Any Mallows-Type Models.**  Chen et al. (2025) studies using Mallows-$\theta$ and Mallows-$\phi$ model to conduct alignment on pairwise preference data. They propose a direct approach to estimate the dispersion parameter $\phi(x)$ without any pretraining or learning.

The idea is to qualitatively relate $\phi(x)$ to the empirical output distribution of the LLM. Intuitively, when $-\log(\phi(x))$ is large, preferences are highly concentrated and the next-token distribution collapses to a point mass, whereas when $-\log(\phi(x))$ approaches zero, the distribution becomes uniform. On the other hand, Shannon's entropy $H(X) = 0$ when $X$ is a point mass, and $H(X) = \log n$ when $X$ is uniform on $n$ points. Motivated by this observation, they propose:

$$-\log\left(H(\pi(\cdot\mid x))/\log n\right),\tag{15}$$

as a proxy to $-\log\phi(x)$, where $\pi(\cdot\mid x)$ can be either the pretrained LM model or the SFT model. Furthermore, they approximate the entropy term in (15) via a realization of a sequence of $N = \max(|Y_w|, |Y_l|)$ tokens $\{Y_w^i, Y_l^i\}_{i=1,\dots,N}$ given the prompt $X$:

$$H(\pi(\cdot\mid X)) \approx \frac{1}{2}\sum_{i=1}^{N-1}\left[H(Y^{i+1}\mid Y^i = Y_w^i) + H(Y^{i+1}\mid Y^i = Y_l^i)\right],\tag{16}$$

which can be easily computed by the logits of the model given the output data. In this case, $n = V^N$, where $V$ is the token size. This is also related to the predictive entropy (Hernández-Lobato et al., 2014; MacKay, 1992) of the next-token predictions.

Finally, the authors validate that this entropy-based estimator closely matches the true dispersion in a synthetic experiment setup.

In this paper, we adopt Chen et al. (2025) to approximate dispersion using Shannon entropy. While their method assumes pairwise comparisons, we extend it to multiple responses. Specifically, for a prompt $X$ with response set $S = \{Y_1, \dots, Y_{|S|}\}$, we approximate $-\log(\phi(X))$ as

$$-\log\left(\tfrac{1}{|S|\log n}\sum_{i=1}^{|S|}\sum_{j=1}^{N-1}H(Y^{j+1}\mid Y^j = Y_i^j)\right),$$

where $H(\cdot|\cdot)$ denotes the conditional Shannon entropy, which can be directly computed from the model's logits. Here, $Y_i^j$ is the $j_{th}$ token of response $i$, $N = \max(|Y_1|, \dots, |Y_{|S|}|)$, and $n = V^N$ with vocabulary size $V$.

**Sigmoid-Smoothed Objectives.**  The objectives in (10), (12), and (14) cannot be directly optimized with gradient-based methods, since they involve step functions that are either discontinuous or flat almost everywhere, yielding zero gradients for most values of $\theta$. To make these objectives amenable

to efficient optimization with preference data, we replace the indicator functions with a sigmoid approximation. This smoothing technique is widely adopted in the machine learning literature (see, e.g., (Qin et al., 2010; Bruch et al., 2019)). Specifically, we approximate $\mathbb{I}\{x < 0\}$ using $\sigma(-x)$. An illustration is provided in Figure 2a.

This design brings two significant benefits:

(i) *Computational benefits*: The original indicator-based objectives are discontinuous at zero, making gradient-based training unstable. The sigmoid function smooths these objectives, facilitating efficient optimization.

(ii) *Soft and robust penalties*: The sigmoid function offers a continuous, differentiable alternative to the indicator, yielding a "soft" loss that reflects the magnitude of preference violations rather than just their direction. For example, in the loss function (10), under the indicator function, only the relative ordering between $\beta \log \frac{\pi_\theta(y_w|x)}{\pi_{\text{ref}}(y_w|x)}$ and $\beta \log \frac{\pi_\theta(y_l|x)}{\pi_{\text{ref}}(y_l|x)}$ matters—i.e., the loss is zero as long as the preferred response scores higher. In contrast, the sigmoid penalty decreases smoothly as the score gap widens, encouraging the model to not only rank preferred responses above dispreferred ones, but to do so confidently. For instance, even if the inequality holds, a small margin will still incur non-trivial loss, while a large margin will yield a smaller loss. This is conceptually similar to incorporating a margin term in the loss function, as seen in prior works (Zhao et al., 2023; Meng et al., 2024; Azar et al., 2024; Hong et al., 2024).

Overall, this smoothing approach allows us to retain the structure of preference-based training while enabling more stable and informative gradient signals during optimization. The final objectives are summarized in Table 1.

# E PROOFS

**Proof of Theorem 1.** The *Multinomial Logit* (MNL) model (McFadden, 1972) is one of the most widely used utility-based discrete choice models. It assumes that each alternative $y_i \in S$ is associated with a latent utility $u_{y_i}(x) = \nu_{y_i}(x) + \epsilon_i$, where $\nu_{y_i}(x)$ is a deterministic component and $\epsilon_i$ follows an independent Gumbel distribution. Under this assumption, the choice probability of selecting alternative $y_w$ from assortment $S$ takes the closed form

$$\mathbb{P}(y_w \mid S ; x) = \frac{\exp(\nu_{y_w}(x))}{\sum_{y_i \in S} \exp(\nu_{y_i}(x))}.$$

By identifying the deterministic utility $\nu_y(x)$ as the reward function defined in (2), the normalization constant $Z(x)$ cancels out, and we are left with:

$$\mathbb{P}_{\pi_\theta}(y_w \mid S ; x) = \frac{e^{\beta \log \frac{\pi_\theta(y_w|x)}{\pi_{\text{ref}}(y_w|x)}}}{\sum_{y_i \in S} e^{\beta \log \frac{\pi_\theta(y_i|x)}{\pi_{\text{ref}}(y_i|x)}}}.$$

Maximizing the likelihood yields the following objective:

$$\min_{\pi_\theta} \; -\mathbb{E}_{(x,S,y_w) \sim \mathcal{D}} \left[ \beta \log \frac{\pi_\theta(y_w|x)}{\pi_{\text{ref}}(y_w|x)} - \log \sum_{y_i \in S} e^{\beta \log \frac{\pi_\theta(y_i|x)}{\pi_{\text{ref}}(y_i|x)}} \right],$$

which is equivalent to the objective in (7). $\square$

**Proof of Theorem 2.** Following Feng & Tang (2023), the probability that an individual selects a top-$k$ choice $\mu^k = y_1 \succ y_2 \succ \ldots \succ y_k$ out of an assortment $S \subseteq \mathcal{N}$ is

$$\mathbb{P}(\mu^k \mid S ; x) = \prod_{i=1}^{k} \frac{e^{\nu_{y_i}(x)}}{\sum_{j=i}^{k} e^{\nu_{y_j}(x)} + \sum_{y_h \in S \setminus \{y_1, \ldots, y_k\}} e^{\nu_{y_h}(x)}}.$$

By identifying the deterministic utility $\nu_y(x)$ as the reward function defined in (2), we have that the optimal RLHF policy satisfies

$$\mathbb{P}_{\pi_\theta}(\mu^k \mid S ; x) = \prod_{i=1}^{k} \frac{e^{\beta \log \frac{\pi_\theta(y_i|x)}{\pi_{\text{ref}}(y_i|x)}}}{\sum_{j=i}^{k} e^{\beta \log \frac{\pi_\theta(y_j|x)}{\pi_{\text{ref}}(y_j|x)}} + \sum_{y_h \in S \setminus \{y_1, \ldots, y_k\}} e^{\beta \log \frac{\pi_\theta(y_h|x)}{\pi_{\text{ref}}(y_h|x)}}}.$$

To maximize the likelihood, our objective becomes:

$$\min_{\pi_\theta} -\mathbb{E}_{(x,S,\mu^k)\sim\mathcal{D}}\left[\sum_{i=1}^{k}\beta\log\frac{\pi_\theta(y_i|x)}{\pi_{\text{ref}}(y_i|x)} - \sum_{i=1}^{k}\log\Big(\sum_{j=i}^{k}e^{\beta\log\frac{\pi_\theta(y_j|x)}{\pi_{\text{ref}}(y_j|x)}} + \sum_{\substack{y_h\in \\ S\setminus\{y_1,\ldots,y_k\}}}e^{\beta\log\frac{\pi_\theta(y_h|x)}{\pi_{\text{ref}}(y_h|x)}}\Big)\right],$$

which is equivalent to the objective in Theorem 2. $\square$

**Proof of Theorem 3.** We consider optimizing the following objective:

$$\max_{\pi_\theta}\mathbb{E}_{x\sim\mathcal{D}}\Big[\mathbb{E}_{y\sim\pi_\theta(y|x)}\big[-\mu_0^{-1}(y\mid x)\big] - \beta\,\text{KL}\big(\pi_\theta(\cdot\mid x)\,\|\,\pi_{\text{ref}}(\cdot\mid x)\big)\Big], \tag{17}$$

As shown in section A.1 of Rafailov et al. (2023), the optimum of such a KL-constrained reward maximization objective has the form of

$$\pi_\theta(y\mid x) = \tfrac{1}{Z(x)}\pi_{\text{ref}}(y\mid x)\exp\Big(-\tfrac{\mu_0^{-1}(y|x)}{\beta}\Big),$$

where $Z(x) = \sum_y\pi_{\text{ref}}(y\mid x)\exp(-\frac{1}{\beta}\mu_0^{-1}(y\mid x))$ is the partition function. By moving terms, we have

$$-\mu_0^{-1}(y\mid x) = \beta\log\frac{\pi_\theta(y|x)}{\pi_{\text{ref}}(y|x)} + \beta\log Z(x). \tag{18}$$

Combining (9) and (18), we have that the optimal RLHF policy $\pi_\theta(\cdot\mid x)$ for (17) satisfies

$$\mathbb{P}_{\pi_\theta}\big(y_w\mid S\,;x\big) = \frac{\phi(x)^{\sum_{y_i\in S\setminus\{y_w\}}\mathbb{I}\{-\beta\log\frac{\pi_\theta(y_w|x)}{\pi_{\text{ref}}(y_w|x)}-\beta\log Z(x)>-\beta\log\frac{\pi_\theta(y_i|x)}{\pi_{\text{ref}}(y_i|x)}-\beta\log Z(x)\}}}{1+\phi(x)+\cdots+\phi(x)^{|S|-1}}.$$

Maximizing the likelihood leads to the following objective:

$$\min_{\pi_\theta} -\mathbb{E}_{(x,S,y_w)\sim\mathcal{D}}\left[\log\frac{\phi(x)^{\sum_{y_i\in S\setminus\{y_w\}}\mathbb{I}\{-\beta\log\frac{\pi_\theta(y_w|x)}{\pi_{\text{ref}}(y_w|x)}>-\beta\log\frac{\pi_\theta(y_i|x)}{\pi_{\text{ref}}(y_i|x)}\}}}{1+\phi(x)+\cdots+\phi(x)^{|S|-1}}\right]$$

$$= \min_{\pi_\theta} -\mathbb{E}_{(x,S,y_w)\sim\mathcal{D}}\left[\sum_{y_i\in S\setminus\{y_w\}}\mathbb{I}\{-\beta\log\tfrac{\pi_\theta(y_w|x)}{\pi_{\text{ref}}(y_w|x)} > -\beta\log\tfrac{\pi_\theta(y_i|x)}{\pi_{\text{ref}}(y_i|x)}\}\log\phi(x) - C(x)\right],$$

where $C(x) = \log(1+\phi(x)+\cdots+\phi(x)^{|S|-1})$ is constant with respect to the policy and thus does not affect the optimal solution. This formulation is equivalent to the objective in (10). $\square$

**Proof of Theorem 4.** Following (11) and a similar discussion as in the proof of Theorem 3, we have the optimal RLHF policy $\pi_\theta(\cdot\mid x)$ for (17) satisfies

$$\mathbb{P}_{\pi_\theta}\big(\mu^k\mid S\,;x\big) = \tfrac{\psi(|S|-k,\phi(x))}{\psi(|S|,\phi(x))}\cdot\phi(x)^{d_{\pi_\theta}(\mu^k,S)},$$

where the exponent term $d(\mu^k,S)$ is

$$d_{\pi_\theta}(\mu^k,S) = \sum_{i=1}^{k-1}\mathbb{I}\left\{-\beta\log\tfrac{\pi_\theta(y_i|x)}{\pi_{\text{ref}}(y_i|x)} > -\beta\log\tfrac{\pi_\theta(y_{i+1}|x)}{\pi_{\text{ref}}(y_{i+1}|x)}\right\}\cdot(|S|-i)+$$

$$\sum_{y_j\in S\setminus\{y_1,\ldots,y_k\}}\mathbb{I}\left\{-\beta\log\tfrac{\pi_\theta(y_k|x)}{\pi_{\text{ref}}(y_k|x)} > -\beta\log\tfrac{\pi_\theta(y_j|x)}{\pi_{\text{ref}}(y_j|x)}\right\}.$$

Maximizing the likelihood leads to the following objective:

$$\min_{\pi_\theta} -\mathbb{E}_{(x,S,\mu^k)\sim\mathcal{D}}\left[\log\big(\tfrac{\psi(|S|-k,\phi(x))}{\psi(|S|,\phi(x))}\cdot\phi(x)^{d_{\pi_\theta}(\mu^k,S)}\big)\right]$$

$$= \min_{\pi_\theta} -\mathbb{E}_{(x,S,\mu^k)\sim\mathcal{D}}\left[\log\tfrac{\psi(|S|-k,\phi(x))}{\psi(|S|,\phi(x))} + d_{\pi_\theta}(\mu^k,S)\log\phi(x)\right],$$

where $\log\frac{\psi(|S|-k,\phi(x))}{\psi(|S|,\phi(x))}$ is constant with respect to the policy and thus does not affect the optimal solution. This formulation is equivalent to the objective in (12). $\square$

**Proof of Theorem 5.** Theorem 5 is a special case of Theorem 3 with $S = \{y_w, y_l\}$. $\square$

# F  GRADIENT ANALYSIS

Let $f_\theta(x, y_1, y_2) := \beta \log \frac{\pi_\theta(y_1|x)}{\pi_{\text{ref}}(y_1|x)} - \beta \log \frac{\pi_\theta(y_2|x)}{\pi_{\text{ref}}(y_2|x)}$ for shorthand notation. We next provide an analysis of various preference optimization models to shed light on their training procedures.

## F.1  MNL

In this section, we derive the gradients of DPO, MNL-PO-Discrete, and MNL-PO-Top-k, and then compare their update mechanisms.

### F.1.1  DPO

Recap the DPO gradient below:

$$\nabla_\theta \mathcal{L}_{\text{DPO}}(\pi_\theta) = -\beta \, \mathbb{E}_{(x,y_w,y_l)\sim\mathcal{D}} \left[ \underbrace{\sigma\big(f_\theta(x, y_l, y_w)\big)}_{\substack{\text{higher weight when} \\ \text{reward estimate is wrong}}} \Big( \underbrace{\nabla_\theta \log \pi_\theta(y_w \mid x)}_{\text{increase likelihood of } y_w} - \underbrace{\nabla_\theta \log \pi_\theta(y_l \mid x)}_{\text{decrease likelihood of } y_l} \Big) \right].$$

### F.1.2  MNL-PO-DISCRETE

The gradient of $\mathcal{L}_{\text{MNL}-\text{PO}-\text{Discrete}}$ with respect to parameters $\theta$ takes the following formulation:

$$\nabla_\theta \mathcal{L}_{\text{MNL}-\text{PO}-\text{Discrete}}(\pi_\theta)$$
$$= -\beta \mathbb{E}\left[ \underbrace{\sigma(\log \sum_{y_i \in S\setminus\{y_w\}} \exp(f_\theta(x, y_i, y_w)))}_{\text{higher weight when reward deviates from preference}} \cdot \Big[ \nabla_\theta \log \pi_\theta(y_w|x) - \sum_{\substack{y_i \in \\ S\setminus\{y_w\}}} \frac{\nabla_\theta \log \pi_\theta(y_i|x)}{\underbrace{\sum_{y_j \in S\setminus\{y_w\}} \exp(f_\theta(x, y_j, y_i))}_{\text{higher weight when reward is larger}}} \Big] \right],$$

where the expectation is with respect to $(x, S, y_w) \sim \mathcal{D}$. As MNL-PO-Discrete is a special case of MNL-PO-Topk with ranking length $k = 1$, we defer its derivation to the next section.

### F.1.3  MNL-PO-TOP-K

The gradient of $\mathcal{L}_{\text{MNL}-\text{PO}-\text{Top}-\text{k}}$ with respect to parameters $\theta$ takes the following formulation:

$$\nabla_\theta \mathcal{L}_{\text{MNL}-\text{PO}-\text{Top}-\text{k}}(\pi_\theta)$$
$$= -\beta \mathbb{E}\Bigg[ \underbrace{\sum_{i=1}^{k}}_{\substack{\text{ranked preference} \\ y_1 \succ \dots \succ y_i | S \,;x}} \underbrace{\sigma\Big( \log \sum_{y_j \in S\setminus\{y_1,\dots,y_i\}} \exp(f_\theta(x, y_j, y_i)) \Big)}_{\text{higher weight when reward deviates from preference}}$$
$$\times \Big[ \nabla_\theta \log \pi_\theta(y_i|x) - \sum_{y_j \in S\setminus\{y_1,\dots,y_i\}} \frac{\nabla_\theta \log \pi_\theta(y_j|x)}{\underbrace{\sum_{y_{j'} \in S\setminus\{y_1,\dots,y_i\}} \exp(f_\theta(x, y_{j'}, y_j))}_{\text{higher weight when reward is larger}}} \Big] \Bigg], \tag{19}$$

where the expectation is with respect to $(x, S, \mu^k) \sim \mathcal{D}$.

The gradient of the MNL-PO-Top-k loss increases the likelihood of the chosen responses while decreasing the likelihood of all unchosen responses. Specifically, each preference relation $(y_1 \succ \dots \succ y_i \mid S \,; x)$ is weighted by the degree to which the implicit reward deviates from the observed preference. Furthermore, MNL-PO-Top-k differentiates among the gradients of unchosen responses: the gradient for an unchosen response $y_j$ is scaled by $\frac{1}{\sum_{y_{j'} \in S\setminus\{y_1,\dots,y_i\}} \exp(f_\theta(x, y_{j'}, y_j))} =$

$\frac{\exp(\beta \log \frac{\pi_\theta(y_j|x)}{\pi_{\text{ref}}(y_j|x)})}{\sum_{y_{j'} \in S\setminus\{y_1,\dots,y_i\}} \exp(\beta \log \frac{\pi_\theta(y'_j|x)}{\pi_{\text{ref}}(y'_j|x)})}$. This factor captures the relative reward of $y_j$ compared with the other unchosen responses.

**Derivation.** Recall that $f_\theta(x, y_1, y_2) = \beta \log \frac{\pi_\theta(y_1|x)}{\pi_{\text{ref}}(y_1|x)} - \beta \log \frac{\pi_\theta(y_2|x)}{\pi_{\text{ref}}(y_2|x)}$. The MNL-PO-Top-k loss takes the following form:

$$\mathcal{L}_{\text{MNL}-\text{PO}-\text{Top}-k}(\pi_\theta) = -\mathbb{E}_{(x,S,\mu^k)\sim\mathcal{D}} \left[ \sum_{i=1}^{k} \log \sigma \Big( -\log \sum_{y_j \in S\setminus\{y_1,...,y_i\}} \exp(f_\theta(x, y_j, y_i)) \Big) \right]$$

The gradient of $f_\theta(x, y_1, y_2)$ can be formulated as:

$$\nabla_\theta f_\theta(x, y_1, y_2) = \beta(\nabla_\theta \log \pi_\theta(y_1|x) - \nabla_\theta \log \pi_\theta(y_2|x)) \tag{20}$$

Using properties of the sigmoid function that $\sigma'(x) = \sigma(x)(1 - \sigma(x)) = \sigma(x)\sigma(-x)$ and thus $((\log \sigma(x))' = \frac{1}{\sigma(x)} \times \sigma(x)\sigma(-x) = \sigma(-x)$, we have:

$\nabla_\theta \mathcal{L}_{\text{MNL}-\text{PO}-\text{Top}-k}(\pi_\theta)$

$= -\mathbb{E} \left[ \sum_{i=1}^{k} \nabla_\theta \log \sigma \left( -\log \sum_{y_j \in S\setminus\{y_1,...,y_i\}} \exp(f_\theta(x, y_j, y_i)) \right) \right]$

$= \mathbb{E} \left[ \sum_{i=1}^{k} \sigma \left( \log \sum_{y_j \in S\setminus\{y_1,...,y_i\}} \exp(f_\theta(x, y_j, y_i)) \right) \cdot \nabla_\theta \log \sum_{y_j \in S\setminus\{y_1,...,y_i\}} \exp(f_\theta(x, y_j, y_i)) \right]$

$((\log \sigma(x))' = \sigma(-x))$

$= \mathbb{E} \left[ \sum_{i=1}^{k} \sigma \left( \log \sum_{y_j \in S\setminus\{y_1,...,y_i\}} \exp(f_\theta(x, y_j, y_i)) \right) \cdot \frac{\sum_{y_j \in S\setminus\{y_1,...,y_i\}} \exp(f_\theta(x,y_j,y_i))\cdot\nabla_\theta f_\theta(x,y_j,y_i)}{\sum_{y_{j'} \in S\setminus\{y_1,...,y_i\}} \exp(f_\theta(x,y_{j'},y_i))} \right]$

$= -\beta\mathbb{E} \left[ \sum_{i=1}^{k} \sigma \left( \log \sum_{y_j \in S\setminus\{y_1,...,y_i\}} \exp(f_\theta(x, y_j, y_i)) \right) \right.$

$\left. \times \sum_{y_j \in S\setminus\{y_1,...,y_i\}} \frac{\nabla_\theta \log \pi_\theta(y_i|x) - \nabla_\theta \log \pi_\theta(y_j|x)}{\sum_{y_{j'} \in S\setminus\{y_1,...,y_i\}} \exp(f_\theta(x,y_{j'},y_i) - f_\theta(x,y_j,y_i))} \right] \tag{Eq. (20)}$

$= -\beta\mathbb{E} \left[ \sum_{i=1}^{k} \sigma \left( \log \sum_{y_j \in S\setminus\{y_1,...,y_i\}} \exp(f_\theta(x, y_j, y_i)) \right) \right.$

$\left. \times \left[ \nabla_\theta \log \pi_\theta(y_i|x) - \sum_{y_j \in S\setminus\{y_1,...,y_i\}} \frac{\nabla_\theta \log \pi_\theta(y_j|x)}{\sum_{y_{j'} \in S\setminus\{y_1,...,y_i\}} \exp(f_\theta(x,y_{j'},y_j))} \right] \right]$

The last equation is because:

$$\sum_{y_j \in S\setminus\{y_1,...,y_i\}} \frac{1}{\sum_{y_{j'} \in S\setminus\{y_1,...,y_i\}} \exp(f_\theta(x,y_{j'},y_i) - f_\theta(x,y_j,y_i))} = \frac{\sum_{y_j \in S\setminus\{y_1,...,y_i\}} \exp(f_\theta(x,y_j,y_i))}{\sum_{y_{j'} \in S\setminus\{y_1,...,y_i\}} \exp(f_\theta(x,y_j',y_i))} = 1, \forall\, i.$$

### F.1.4 COMPARISON OF GRADIENT UPDATES

**DPO** updates the model by increasing the likelihood of the preferred response $y_w$ and decreasing that of the dispreferred response $y_l$. The update weight is larger when the model's reward estimate disagrees with the preference. This ensures that learning focuses on correcting mistakes, especially in cases where the model is misaligned with the observed preference.

**MNL-PO-Discrete** compares the preferred response $y_w$ against all other alternatives $y_i \in S \setminus \{y_w\}$. The update weight grows when the reward deviates from the preference, and it is further adjusted according to the relative reward magnitudes across the choice set. As a result, the gradient not only enforces the winner against individual competitors but also reflects the overall reward distribution of the choice set.

**MNL-PO-Top-k** extends this to ranked preferences $y_1 \succ \ldots \succ y_k \mid S$. The gradient sequentially enforces each ranking position by comparing $y_i$, $i = 1, \ldots, k$ against the remaining alternatives.

Similar to MNL-PO-Discrete, the update is stronger when the reward diverges from the observed preference and when the competitor's reward is larger. This design enables the model to learn the entire preference ranking rather than only the top choice, sharpening distinctions across multiple ranking positions.

## F.2 MALLOWS-RMJ

In this section, we derive the gradients of Mallows-RMJ-PO-Pairwise, Mallows-RMJ-PO-Discrete, and Mallows-RMJ-PO-Top-k, and then compare their update mechanisms.

### F.2.1 MALLOWS-RMJ-PO-PAIRWISE

The Mallows-RMJ-PO-Pairwise loss takes the following form:

$$\mathcal{L}_{\text{Mallows}-\text{RMJ}-\text{PO}-\text{Pairwise}}(\pi_\theta) = -\mathbb{E}_{(x,y_w,y_l)\sim\mathcal{D}}\left[\log\phi(x)\cdot\sigma(f_\theta(x,y_l,y_w))\right].$$

The gradient of $\mathcal{L}_{\text{Mallows}-\text{RMJ}-\text{PO}-\text{Pairwise}}$ with respect to parameters $\theta$ takes the following formulation:

$$\nabla_\theta\mathcal{L}_{\text{Mallows}-\text{RMJ}-\text{PO}-\text{Pairwise}}(\pi_\theta)$$
$$= -\mathbb{E}\left[\log\phi(x)\ \sigma'\big(f_\theta(x,y_l,y_w)\big)\ \nabla_\theta f_\theta(x,y_l,y_w)\right]$$
$$= \beta\mathbb{E}\Bigg[\underbrace{-\log\phi(x)}_{\substack{\text{larger weight}\\\text{for less dispersed}}}\ \underbrace{\sigma\big(f_\theta(x,y_l,y_w)\big)\big(1-\sigma\big(f_\theta(x,y_l,y_w)\big)\big)}_{\text{larger weight when reward estimates are closer}}\cdot\Big[\underbrace{\nabla_\theta\log\pi_\theta(y_l|x)}_{\text{decrease likelihood of }y_l}-\underbrace{\nabla_\theta\log\pi_\theta(y_w|x)}_{\text{increase likelihood of }y_w}\Big]\Bigg],$$

where the expectation is with respect to $(x,y_w,y_l)\sim\mathcal{D}$. See Figure 2b for an illustration of $\sigma'(\cdot)$.

### F.2.2 MALLOWS-RMJ-PO-DISCRETE

The Mallows-RMJ-PO-Discrete loss takes the following form:

$$\mathcal{L}_{\text{Mallows}-\text{RMJ}-\text{PO}-\text{Discrete}}(\pi_\theta) = -\mathbb{E}_{(x,S,y_w)\sim\mathcal{D}}\log\phi(x)\cdot\sum_{y_i\in S\setminus\{y_w\}}\sigma\big(f_\theta(x,y_i,y_w)\big).$$

The gradient of $\mathcal{L}_{\text{Mallows}-\text{RMJ}-\text{PO}-\text{Discrete}}$ with respect to parameters $\theta$ takes the following formulation:

$$\nabla_\theta\mathcal{L}_{\text{Mallows}-\text{RMJ}-\text{PO}-\text{Discrete}}(\pi_\theta)$$
$$= -\mathbb{E}\Bigg[\log\phi(x)\sum_{y_i\in S\setminus\{y_w\}}\sigma'\big(f_\theta(x,y_i,y_w)\big)\ \nabla_\theta f_\theta(x,y_i,y_w)\Bigg]$$
$$= \beta\mathbb{E}\Bigg[\underbrace{-\log\phi(x)}_{\text{larger weight for less dispersed}}\sum_{y_i\in S\setminus\{y_w\}}\underbrace{\sigma\big(f_\theta(x,y_i,y_w)\big)\big(1-\sigma\big(f_\theta(x,y_i,y_w)\big)\big)}_{\text{larger weight when reward estimates are closer}}\cdot$$
$$\Big[\underbrace{\nabla_\theta\log\pi_\theta(y_i|x)}_{\text{decrease likelihood of }y_i}-\underbrace{\nabla_\theta\log\pi_\theta(y_w|x)}_{\text{increase likelihood of }y_w}\Big]\Bigg],$$

where the expectation is with respect to $(x,S,y_w)\sim\mathcal{D}$.

### F.2.3 MALLOWS-RMJ-PO-TOP-K

The Mallows-RMJ-PO-Top-k loss takes the following form:

$$\mathcal{L}_{\text{Mallows}-\text{RMJ}-\text{PO}-\text{Top}-\text{k}}(\pi_\theta) =$$
$$- \mathbb{E}_{(x,S,\mu^k)\sim\mathcal{D}}\Bigg[\log\phi(x)\Big(\sum_{i=1}^{k-1}(|S|-i)\,\sigma(f_\theta(x,y_{i+1},y_i)) + \sum_{y_j\in S\setminus\{y_1,\dots,y_k\}}\sigma(f_\theta(x,y_j,y_k))\Big)\Bigg].$$

The gradient of $\mathcal{L}_{\text{Mallows}-\text{RMJ}-\text{PO}-\text{Top}-\text{k}}$ with respect to parameters $\theta$ takes the following formulation:

$$\nabla_\theta \mathcal{L}_{\text{Mallows}-\text{RMJ}-\text{PO}-\text{Top}-\text{k}}(\pi_\theta)$$

$$= -\mathbb{E}\Bigg[ \log\phi(x) \Big( \sum_{i=1}^{k-1} (|S|-i)\,\sigma'(f_\theta(x,y_{i+1},y_i))\,\nabla_\theta f_\theta(x,y_{i+1},y_i) $$

$$+ \sum_{y_j \in S\setminus\{y_1,\ldots,y_k\}} \sigma'(f_\theta(x,y_j,y_k))\,\nabla_\theta f_\theta(x,y_j,y_k) \Big) \Bigg]$$

$$= \beta\mathbb{E}\Bigg[ \underbrace{-\log\phi(x)}_{\substack{\text{larger weight for}\\\text{less dispersed}}} \Bigg( \sum_{i=1}^{k-1} \underbrace{(|S|-i)}_{\text{larger weight for top position}} \underbrace{\sigma\big(f_\theta(x,y_{i+1},y_i)\big)\big(1-\sigma\big(f_\theta(x,y_{i+1},y_i)\big)\big)}_{\text{larger weight when reward estimates are closer}}$$

$$\times \big( \underbrace{\nabla_\theta \log\pi_\theta(y_{i+1}\mid x)}_{\text{decrease likelihood of } y_{i+1}} - \underbrace{\nabla_\theta \log\pi_\theta(y_i\mid x)}_{\text{increase likelihood of } y_i} \big)$$

$$+ \sum_{\substack{y_j \in \\ S\setminus\{y_1,\ldots,y_k\}}} \underbrace{\sigma\big(f_\theta(x,y_j,y_k)\big)\big(1-\sigma\big(f_\theta(x,y_j,y_k)\big)\big)}_{\text{larger weight when reward estimates are closer}} \big( \underbrace{\nabla_\theta \log\pi_\theta(y_j\mid x)}_{\text{decrease likelihood of } y_j} - \underbrace{\nabla_\theta \log\pi_\theta(y_k\mid x)}_{\text{increase likelihood of } y_k} \big) \Bigg) \Bigg],$$

where the expectation is with respect to $(x, S, \mu^k) \sim \mathcal{D}$.

### F.2.4 COMPARISON OF GRADIENT UPDATES

**Mallows-RMJ-PO-Pairwise** focuses on pairwise comparisons between a preferred response $y_w$ and a dispreferred response $y_l$. The gradient increases the likelihood of $y_w$ while decreasing that of $y_l$, with stronger updates when preferences are less dispersed and when the two responses have similar rewards. This concentrates learning on the harder, more informative comparisons.

**Mallows-RMJ-PO-Discrete** generalizes this idea to a set of responses. The update pushes up the likelihood of the preferred response against all alternatives in $S \setminus \{y_w\}$. Again, the updates are stronger when preferences are concentrated and when the competing responses are close in reward, ensuring sharper separation between the winner and its competitors.

**Mallows-RMJ-PO-Top-k** extends to ranked choice feedback. Here the gradient simultaneously enforces the full ranking: higher-ranked responses are promoted while lower-ranked ones are suppressed. The update is amplified for top positions, for less dispersed preferences, and for cases where neighboring rewards are close. This design emphasizes both the most critical ranking positions and the harder comparisons, yielding sharper alignment to the preference order.

## G EXPERIMENTAL DETAILS

Our method is implemented by modifying the `DPO Trainer` and `DPO Config` in the `TRL` library.

**Computation Environment** All training runs were performed in Python 3.10.16 with PyTorch 2.6.0 on a server with 7 NVIDIA H100 GPUs each with 80 GB of memory, equipped with Ubuntu 22.04.2 LTS. The response-generation step in evaluations is performed on an NVIDIA Quadro RTX 6000 GPU with 24 GB of memory.

**Dataset** We provide descriptions of the UltraFeedback train dataset in Table 6. We construct a ranking-based preference set for ranked choice training in the following way. We generate five distinct responses for each prompt in the UltraFeedback dataset (Cui et al., 2023) using a sampling temperature of 0.8. We then score and rank these five responses with the Skywork-Reward-V2-Llama-3.1-8B reward model. In the pairwise setup, we adopt the literature's tradition and use the top- and bottom- ranked responses as the assortment. For ranked choice training, we truncate each full ranking to generate the required data.

The UltraFeedback test dataset contains 1961 prompts, and we use the chosen responses as the baseline. This dataset serves as an in-distribution test.

**Benchmarks**  We provide descriptions of the evaluation benchmarks in Table 7.

**Open Source Models**  The Hugging Face IDs of the base models and reward model used in our experiments are listed in Table 5.

Table 5: Base Models and Reward Models Used in Experiments.

| Model | Hugging Face ID |
|---|---|
| Llama-3-8B-Instruct | meta-llama/Meta-Llama-3-8B-Instruct |
| Gemma-2-9B-it | google/gemma-2-9b-it |
| Mistral-7B-Instruct | mistralai/Mistral-7B-Instruct-v0.2 |
| Skywork-Reward-V2-Llama-3.1-8B | Skywork/Skywork-Reward-V2-Llama-3.1-8B |

Table 6: Overview of the UltraFeedback Train Dataset (Cui et al., 2023).

| Item | Description |
|---|---|
| **Data Scale** | ∼64K prompts |
| **Prompt Sources** | UltraChat, ShareGPT, Evol-Instruct, TruthfulQA, FalseQA, FLAN |
| **Prompt Types** | Open-ended dialogue
Instruction-following (writing, coding, summarization, translation)
Factual QA, adversarial/false QA
Standard NLP tasks (classification, reasoning, reading comprehension) |
| **Response Source** | Generated by base model |
| **Ranking Method** | Scored by Skywork-V2 (Skywork-Reward-V2-Llama-3.1-8B) |

Table 7: Evaluation details for AlpacaEval 2 (Dubois et al., 2024) and Arena-Hard (Li et al., 2024).

| | # Exs. | Baseline Model | Judge Model | Scoring Type | Metric |
|---|---|---|---|---|---|
| **AlpacaEval 2** | 805 | GPT-4-Turbo | GPT-4.1-mini | Pairwise Comparison | LC & Win Rate |
| **Arena-Hard** | 500 | GPT-4-0314 | GPT-4.1-mini | Pairwise Comparison | Win Rate |

**Notes:** The baseline model refers to the model compared against. GPT-4 Turbo corresponds to GPT-4-Preview-1106. GPT-4.1-mini corresponds to GPT-4.1-mini-2025-04-14.

### G.1 TRAINING HYPER-PARAMETER TUNING

By default of TRL, we use adamw_torch optimizer.

**Llama-3-8B-Instruct**  We adopt a global batch size of 112, a maximum sequence length of 4096, and a cosine learning rate schedule for one epoch across all training settings. We retrieve the latest models of the baseline methods; their Huggingface IDs are listed in Table 8. The hyperparameters we used for training are in Table 9.

**Gemma-2-9B-it**  We adopt a global batch size of 112, a maximum sequence length of 2048, and a cosine learning rate schedule for one epoch. We retrieve the latest models of the baseline methods; their Huggingface IDs are listed in Table 10. The hyperparameters we used for training are in Table 11.

Table 8: List of Baseline Models Used in Experiments on Llama-3-8B-instruct.

| Baseline Method | Huggingface ID |
|---|---|
| SimPO (Meng et al., 2024) | princeton-nlp/Llama-3-Instruct-8B-SimPO-v0.2 |
| DPO (Rafailov et al., 2023) | princeton-nlp/Llama-3-Instruct-8B-DPO-v0.2 |
| RDPO (Park et al., 2024) | princeton-nlp/Llama-3-Instruct-8B-RDPO-v0.2 |
| CPO (Xu et al., 2024) | princeton-nlp/Llama-3-Instruct-8B-CPO-v0.2 |
| IPO Azar et al. (2024) | princeton-nlp/Llama-3-Instruct-8B-IPO-v0.2 |
| ORPO (Hong et al., 2024) | princeton-nlp/Llama-3-Instruct-8B-ORPO-v0.2 |
| RRHF (Yuan et al., 2023) | princeton-nlp/Llama-3-Instruct-8B-RRHF-v0.2 |
| SLiC-HF (Zhao et al., 2023) | princeton-nlp/Llama-3-Instruct-8B-SLiC-HF-v0.2 |
| KTO (Ethayarajh et al., 2024) | princeton-nlp/Llama-3-Instruct-8B-KTO-v0.2 |

Table 9: Hyperparameters for training on Llama-3-8B-Instruct.

| Method | $\beta$ | Learning Rate | Warmup |
|---|---|---|---|
| Mallows-RMJ-PO-Pairwise | 0.03 | $3 \times 10^{-7}$ | 0.2 |
| MNL-PO-Discrete | 0.01 | $5 \times 10^{-7}$ | 0.1 |
| Mallows-RMJ-PO-Discrete | 0.03 | $3 \times 10^{-7}$ | 0.2 |
| MNL-PO-Top-2 | 0.01 | $5 \times 10^{-7}$ | 0.1 |
| Mallows-RMJ-PO-Top-2 | 0.01 | $3 \times 10^{-7}$ | 0.1 |

Table 10: List of baseline models used in experiments on Gemma-2-9B-it.

| Baseline Method | Huggingface ID |
|---|---|
| SimPO (Meng et al., 2024) | princeton-nlp/gemma-2-9b-it-SimPO |
| DPO (Rafailov et al., 2023) | princeton-nlp/gemma-2-9b-it-DPO |

Table 11: Hyperparameters for training on Gemma-2-9B-it.

| Method | $\beta$ | Learning Rate | Warmup |
|---|---|---|---|
| Mallows-RMJ-PO-Top-2 | 0.01 | $5 \times 10^{-7}$ | 0.2 |

Table 12: List of baseline models used in experiments on Mistral-7B-Instruct.

| Baseline Method | Huggingface ID |
|---|---|
| SimPO (Meng et al., 2024) | princeton-nlp/Mistral-7B-Instruct-SimPO |
| DPO (Rafailov et al., 2023) | princeton-nlp/Mistral-7B-Instruct-DPO |

**Mistral-7B-Instruct** We adopt a global batch size of 112, a maximum sequence length of 2048, and a cosine learning rate schedule for one epoch. We retrieve the latest models of the baseline methods; their Huggingface IDs are listed in Table 12. The hyperparameters we used for training are in Table 13.

Table 13: Hyperparameters for training on Mistral-7B-Instruct.

| Method | $\beta$ | Learning Rate | Warmup |
|---|---|---|---|
| Mallows-RMJ-PO-Top-2 | 0.05 | $5 \times 10^{-7}$ | 0.2 |

## G.2 DECODING HYPERPARAMETERS

For AlpacaEval 2.0, we adopt the default template for evaluators provided by AlpacaEval. For Llama-3-8B-Instruct settings, we adopt the following fixed generation config: max new tokens 4096, temperature 0.7 and top-$p$ 0.1. For Gemma-2-9B-it settings, we adopt the following fixed generation

config: max new tokens 4096, temperature $0.5$ and top-$p$ 1.0. For Arena-Hard-v0.1, we use the default greedy decoding for all settings and methods. For Ultrafeedback, we use the same configurations as those in AlpacaEval 2.0.

## G.3 TRAINING TIME OF MNL AND MALLOWS-RMJ

Table 14: Training Time (Hours) of MNL and Mallows-RMJ.

| Feedback | MNL | Mallows-RMJ |
|---|---|---|
| Pairwise Comparison | 1.5 | 1.5 |
| Discrete Choice | 3.5 | 3.5 |
| Top 2 Choice | 3.5 | 3.5 |

## G.4 ROBUSTNESS CHECK

To enhance cross-judge robustness, we additionally employ GPT-5-mini-2025-08-07 as the judge on Arena-Hard. We report the win rate and 95% confidence interval. The results are shown in Table 15.

Table 15: Evaluation Results for Llama-3-8B-Instruct and Gemma-2-9B-it (Judged by GPT-5-mini).

| Base | Method | WR (%) | 95% CI |
|---|---|---|---|
| | Base Model | 20.9 | [19.3, 22.7] |
| | CPO (Xu et al., 2024) | 23.7 | [22.1, 25.4] |
| | IPO (Azar et al., 2024) | 23.2 | [21.6, 24.7] |
| | ORPO (Hong et al., 2024) | 22.1 | [20.5, 23.8] |
| | RRHF (Yuan et al., 2023) | 22.3 | [20.7, 23.8] |
| | SLiC-HF (Zhao et al., 2023) | 23.1 | [21.5, 24.7] |
| Llama-3-8B-Instruct | KTO (Ethayarajh et al., 2024) | 21.1 | [19.4, 22.5] |
| | DPO (Rafailov et al., 2023) | 25.9 | [24.7, 27.3] |
| | R-DPO (Park et al., 2024) | 24.9 | [22.8, 26.4] |
| | SimPO (Meng et al., 2024) | 27.2 | [25.4, 28.7] |
| | Mallows-RMJ-PO-Pairwise | **27.4** | [25.8, 28.9] |
| | MNL-PO-Discrete | 25.1 | [23.1, 27.0] |
| | Mallows-RMJ-PO-Discrete | 25.4 | [23.8, 27.3] |
| | MNL-PO-Top-2 | 24.9 | [23.6, 26.5] |
| | Mallows-RMJ-PO-Top-2 | 25.8 | [24.1, 27.5] |
| | Base Model | 38.2 | [35.9, 40.3] |
| Gemma-2-9B-it | SimPO (Meng et al., 2024) | 47.0 | [45.3, 49.0] |
| | DPO (Rafailov et al., 2023) | 49.4 | [47.3, 51.6] |
| | Mallows-RMJ-PO-Top-2 | **51.0** | [48.7, 52.8] |

## G.5 MORE ABLATION STUDIES

**Sample Efficiency of Ranked Choice Training Method.** We have evaluated the generalization performance as the amount of training data increases. For reference, when using all training data, the LC and WR for DPO are 41.24 and 40.24, respectively. As shown in Figure 3, Mallows-RMJ-PO-Top-2 can achieve comparable performance to full-data DPO when trained on much smaller data. Notably, strong performance can already be observed when using roughly 40% of the full dataset. This shows that our ranked choice method can achieve high sample efficiency.

**Impact of $\beta$ on generalization performance.** Figure 4 shows the $\beta$ sensitivity.

**Impact of different treatments for ranked responses.** Given a set of ranked responses, one can (i) use only the top–bottom pair (Meng et al., 2024; Chen et al., 2025; Zhao et al., 2024; Gupta et al., 2025), (ii) decompose the ranking into all possible pairwise comparisons (Ouyang et al., 2022; Zhang et al., 2024), or (iii) directly train on ranked choice data (ours). We analyze how these different treatments affect both generalization performance and training efficiency.

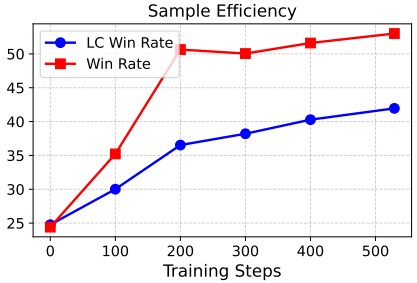

Figure 3: AlpacaEval 2 metrics evolution with training steps for Mallows-RMJ-PO-Top-2.

Figure 4: Effect of tuning $\beta$ for Mallows-RMJ-PO-Top-2.

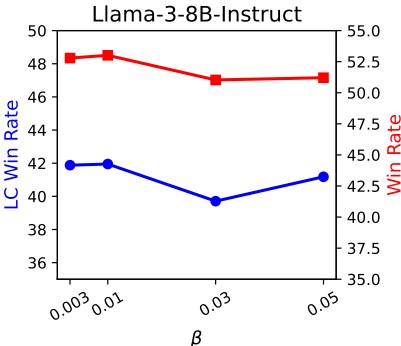

As shown in Table 16, our methods achieve the best generalization performance with only a modest increase in training time. In contrast, decomposing the ranking into pairs greatly inflates the data size, resulting in much longer training time while yielding the worst performance.

Table 16: Evaluation Results of Different Treatments for Ranked Responses.

| Method | AlpacaEval 2 LC (%) | AlpacaEval 2 WR (%) | Arena-Hard WR (%) | Training Time (Hours) |
|---|---|---|---|---|
| DPO | 41.24 (0.35) | 40.24 (1.66) | 32.6 (30.6, 34.7) | 1.5 |
| DPO-AllPairs | 33.02 (0.24) | 38.47 (1.65) | 29.6 (27.3, 31.5) | 11.5 |
| MNL-PO-Top-2 | 40.12 (0.22) | 47.69 (1.67) | 35.8 (33.2, 38.2) | 3.5 |
| Mallows-RMJ-PO-Top-2 | **41.95** (0.26) | **53.01** (1.68) | **37.2** (35.0, 39.4) | 3.5 |

**Notes:** DPO and DPO-AllPairs correspond to treatment (i) and (ii), respectively. Standard errors (in parentheses) follow AlpacaEval 2 metrics, and 95% confidence intervals follow Arena-Hard WR.

## H  QUALITATIVE EXAMPLES

In this section, we present a series of examples for direct comparisons between our RCPO and benchmarks, as shown in Table 17–21. These tables showcase the qualitative examples of model responses.

Table 17: Qualitative Example (SAT Word)

| | |
|---|---|
| **Question:** *Find the answer that best describes the underlined SAT word. Select the correct option and explain the meaning of the underlined word.* | |
| *Despite the cacophony, the student tried to study.* | |
| *A. Loud sounds   B. Difficult subject   C. Late hour   D. Low lighting* | |

| Mallows-RMJ-PO-Top-2 | DPO |
|---|---|
| **The correct answer is A. Loud sounds.** | **The correct answer is A. Loud sounds.** |
| The word "*cacophony*" refers to a harsh, discordant sound or mixture of sounds, often unpleasant to the ear. In this context, "despite the cacophony" means that despite the loud and unpleasant noise, the student tried to study. | The word "*cacophony*" refers to a harsh, discordant sound or a mixture of loud, unpleasant noises. In this context, "despite the cacophony" means that despite the loud and unpleasant sounds (e.g., noise from construction, traffic, etc.) surrounding the student, they still tried to study. |
| **Example:** The construction site outside my window created a cacophony of hammering, drilling, and shouting, but I managed to focus on my work. | **Example:** If a student is trying to study in a busy city street or a noisy café, the cacophony of sounds from cars honking, people talking, and music playing might be distracting, but the student is trying to focus despite these unpleasant sounds. |
| The other options don't fit the context: | |
| **B. Difficult subject:** The word "cacophony" doesn't relate to a challenging academic subject. | |
| **C. Late hour:** While it's possible to study at a late hour, the word "cacophony" doesn't describe the time of day. | |
| **D. Low lighting:** This option is unrelated to the concept of loud, unpleasant sounds. | |

**Notes:** Compared with DPO, the response generated by Mallows-RMJ-PO-Top-2 also explains why the other options are incorrect.

Table 18: Qualitative Example (Excel Test)

| | |
|---|---|
| **Question:** *Please write the Excel function name associated with each description.* | |
| • *Returns the number of days between two dates* | |
| • *Returns the starting position of a text string within another text string* | |
| • *Returns the number in the middle of the set of given numbers* | |

| Mallows-RMJ-PO-Top-2 | DPO |
|---|---|
| Here are the Excel function names associated with each description: | Here are the Excel function names associated with each description: |
| 1. **Returns the number of days between two dates:** `DATEDIF` or `DAYS`
   • `DATEDIF` is a more flexible function that can calculate the difference between two dates in various units (days, months, years, etc.), while `DAYS` is a simpler function that returns the exact number of days between two dates. | 1. **Returns the number of days between two dates:** `DATEDIF` or `DAYS`
However, if you want a single function that returns the exact number of days between two dates, you can use a custom formula: `=A2-A1+1` (assuming A1 and A2 are the start and end dates). |
| 2. **Returns the starting position of a text string within another text string:** `FIND` or `SEARCH`
   • `FIND` is case-sensitive, while `SEARCH` is not. | 2. **Returns the starting position of a text string within another text string:** `FIND` or `SEARCH`
   • `FIND` is case-sensitive, while `SEARCH` is not. |
| 3. **Returns the number in the middle of the set of given numbers:** `MEDIAN`
   • This function returns the middle value in a set of numbers, assuming the set is sorted in ascending order. If the set has an even number of values, `MEDIAN` returns the average of the two middle values. | 3. **Returns the number in the middle of the set of given numbers:** `AVERAGE` or `MEDIAN`
   • If you want to return the exact middle value (in case of an even number of values), you can use `=AVERAGE(MIDIF(A1:A10,1, COUNT(A1:A10)))` (assuming A1:A10 is the range of numbers), where `MIDIF` is a custom array formula.
   • Alternatively, you can use `=INDEX(A1:A10,MATCH(2,IF(A1:A10 <MIN(A1:A10)+MAX(A1:A10), ROW(A1:A10),"")),0))` (assuming A1:A10 is the range of numbers).
   • For a simpler approach, you can use `=AVERAGE(A1:A10)` and then round to the nearest integer if needed.
   • For a more robust approach, use `=MEDIAN(A1:A10)`. |

**Notes:** In the Excel test, Mallows-RMJ-PO-Top-2 answered all three questions correctly, whereas DPO incorrectly calculated the median in the last question using the average function.

Table 19: Qualitative Example (Open Question)

**Question:** *Which libraries are the best for developing deep learning scripts in Python?*

| Mallows-RMJ-PO-Top-2 | DPO |
|---|---|

For developing deep learning scripts in Python, you'll want to focus on libraries that provide efficient implementations of neural networks, automatic differentiation, and optimization algorithms. Here are some of the most popular and widely-used libraries:

1. **TensorFlow (TF)**: An open-source library developed by Google. Provides a Python API, supports both CPU and GPU acceleration, and has a large community of developers.
**Pros:** Wide range of pre-built estimators, flexible architecture, good documentation.
**Cons:** Steeper learning curve, can be slow for small models.

2. **PyTorch**: Known for its dynamic computation graph and automatic differentiation. Popular among researchers who prefer a more Pythonic API.
**Pros:** Easier to learn, dynamic computation graph, fast prototyping.
**Cons:** Less mature than TensorFlow, fewer pre-built estimators.

3. **Keras**: High-level neural networks API written in Python. Can run on top of TensorFlow, PyTorch, or Theano.
**Pros:** Easy to use, rapid prototyping, supports multiple backends.
**Cons:** Limited control over low-level details, not suitable for complex models.

4. **Theano**: Library from the University of Montreal, focused on automatic differentiation and symbolic math.
**Pros:** Strong support for symbolic math, research-friendly, high-performance.
**Cons:** Steep learning curve, less user-friendly.

5. **Microsoft Cognitive Toolkit (CNTK)**: A commercial-grade DL library from Microsoft, known for high-performance computing and large-scale support.
**Pros:** High-performance, supports large-scale models, good documentation.
**Cons:** Commercial license required for some features, less widely used.

6. **Hugging Face Transformers**: Focused on NLP and transformer-based models, provides pre-trained models and simple interfaces.
**Pros:** Easy to use, large community, pre-trained models.
**Cons:** Limited to NLP and transformer-based models.

**When choosing a library, consider:**
* Your project's specific requirements (large datasets/complex models: TensorFlow, PyTorch; simple/rapid prototyping: Keras, Hugging Face).
* Your team's expertise (familiarity makes adoption easier).
* Computational resources (if limited, Keras or Hugging Face can run on CPU/GPU).
* Community support (active communities provide better docs/tutorials).

Ultimately, it's a good idea to experiment with different libraries to determine which best fits your needs and preferences. You can also use libraries like **scikit-learn**, which provides a unified interface for various ML algorithms, including some DL techniques.

---

For developing deep learning scripts in Python, you'll want to focus on libraries that provide efficient implementations of neural networks and related algorithms. Here are some of the most popular and widely used libraries:

1. **TensorFlow (TF)**: An open-source library developed by Google. Popular for deep learning, especially for large-scale projects. Provides a wide range of tools and APIs for building and training neural networks.

2. **PyTorch**: Another open-source library, developed by Facebook's AI Research Lab (FAIR). Known for its dynamic computation graph, automatic differentiation, and ease of use. Particularly popular among researchers and developers who prefer a Pythonic API.

3. **Keras**: A high-level neural networks API written in Python, capable of running on top of TensorFlow, PyTorch, or Theano. Provides an easy-to-use interface for building and experimenting with deep learning models, suitable for beginners and experts.

4. **SciPy**: While not exclusively a deep learning library, SciPy's `scipy.optimize` and `scipy.signal` modules provide useful tools for optimization and signal processing, which are essential for many deep learning algorithms.

For specific tasks, you may also want to consider the following libraries:

5. **OpenCV (Computer Vision)**: For tasks such as image and video processing, feature extraction, and object detection.

6. **NLTK (Natural Language Toolkit)** and **spaCy**: For natural language processing (NLP) tasks, such as text processing, tokenization, and language modeling.

7. **PyCaffe (optional)**: A Python wrapper for the Caffe deep learning framework, useful for deploying pre-trained models.

**When choosing a library, consider the following factors:**
* **Ease of use:** If you're new to deep learning, Keras or PyTorch might be a better choice due to their more Pythonic APIs.
* **Performance:** TensorFlow and PyTorch are generally more performant for large-scale deep learning tasks.
* **Specific requirements:** If you need to work with computer vision or NLP, OpenCV and NLTK/spaCy might be more suitable.
* **Community support:** All of these libraries have large communities and extensive documentation, but TensorFlow and PyTorch have a more extensive ecosystem.

Here's a rough guideline:
• For beginners: Keras or PyTorch.
• For large-scale deep learning: TensorFlow or PyTorch.
• For specific tasks (computer vision, NLP): OpenCV, NLTK/spaCy, or library-specific options (e.g., OpenCV for vision).

Remember, you can always use multiple libraries together to achieve your goals. Happy deep learning!

**Notes:** The response generated by Mallows-RMJ-PO-Top-2 gives more details of pros and cons when listing the options compared to that generated by DPO.

Table 20: Qualitative Example (Factual Explanatory Question)

| **Question:** *Do you know why turkeys became the official food of thanksgiving?* | |
|---|---|
| **Mallows-RMJ-PO-Top-2** | **DPO** |

| | |
|---|---|
| The origins of turkey as the centerpiece of the modern American Thanksgiving meal are not entirely clear, but there are several theories. Here's a brief history: 

 **The Wampanoag and the Pilgrims**: The first Thanksgiving, which took place in 1621, was a harvest feast celebrated by the Pilgrims, early European settlers of Plymouth Colony, and the Wampanoag Native American tribe. The Wampanoag had helped the Pilgrims adapt to the new land and taught them how to farm. The feast likely featured foods native to the region, such as venison, corn, squash, and beans. There is no direct evidence that turkey was served at this specific feast. 

 **Wild Turkeys in North America**: Wild turkeys (*Meleagris gallopavo*) were abundant in North America, and Native Americans had hunted them for centuries. The Pilgrims would have been familiar with wild turkeys, which were a common food source in the region. 

 **17th-Century European Traditions**: In Europe, especially in England, roasted capon (a castrated male chicken) or goose was a traditional centerpiece for special occasions like harvest festivals and celebrations. When European settlers arrived in North America, they brought these culinary traditions with them. 

 **18th and 19th Centuries: The Rise of Turkey as a Symbol of Thanksgiving**: Over time, turkey became a popular choice for special occasions in America, including Thanksgiving. Several factors contributed to its rise: 

 1. **Availability**: Domesticated turkeys were introduced to North America in the early 1600s and became widely available in the 18th century. 

 2. **Marketing and Promotion**: In the mid-1800s, turkey farmers and butchers began promoting turkey as a centerpiece for Thanksgiving, capitalizing on its association with the holiday. 

 3. **Sarah Josepha Hale's Influence**: Sarah Josepha Hale, a prominent American writer and editor, campaigned for a national Thanksgiving holiday through her magazine, *Godey's Lady's Book*. In 1827, she published a recipe for roasted turkey with stuffing, which helped popularize the dish. 

 4. **Abraham Lincoln's Proclamation**: In 1863, during the Civil War, President Abraham Lincoln declared Thanksgiving a national holiday, and turkey became an integral part of the celebration. His wife, Mary Todd Lincoln, was known to serve roasted turkey at the White House. 

 By the early 20th century, turkey had become the traditional main course for Thanksgiving in the United States, largely due to a combination of these factors. While it's unlikely that turkey was the primary protein at the first Thanksgiving, it has since become an integral part of the holiday's culinary tradition. | A common misconception! Turkeys didn't actually become the official food of Thanksgiving because of some historical decree. Instead, wild turkeys were a natural part of the traditional diet and celebrations of early American settlers, particularly the Pilgrims and Wampanoag Native Americans. Here's the backstory: 

 In 1621, the Pilgrims, who had established the Plymouth Colony, held a harvest feast to celebrate their first successful crop yield. They invited the Wampanoag tribe, who had helped them adapt to the new land and taught them agricultural techniques. The feast, now known as the First Thanksgiving, lasted for three days and featured a variety of foods, including: 

 1. Venison (deer meat) 

 2. Wild fowl (likely including ducks, geese, and passenger pigeons) 

 3. corn (maize) 

 4. Squash 

 5. Beans 

 6. Seafood (like cod and eel) 

 Turkeys were present, but not the centerpiece. Wild turkeys were abundant in the region, and they would have been a natural addition to the harvest meal, along with other game meats. 

 Over time, as European customs merged with American traditions, roasted turkey became a staple of annual harvest celebrations, including Thanksgiving, which was not officially declared a national holiday until 1863 by President Abraham Lincoln. By then, turkey had already become a popular choice due to its availability, affordability, and symbolic association with abundance and gratitude. 

 So, while turkeys weren't specifically designated as the "official food" of Thanksgiving, their presence at the First Thanksgiving and their subsequent popularity in American cuisine cemented their place as a classic Thanksgiving dish. |

**Notes:** These outputs are illustrative rather than fact-checked. Mallows-RMJ-PO-Top-2 responds by providing a historical narrative aligned with the question's framing, whereas DPO places greater emphasis on questioning and qualifying the premise of the question.

Table 21: Qualitative Example (Simplification)

**Question:** *Simplify "Most of the basic functions of current virtualization software are inherited from this early VM OS. The virtualization software is now represented with the term hypervisor. Hardware virtualization is used to share the resources of the cloud providers effectively with the customers by generating an illusion of dedicated computing, storage, and networking on a computing infrastructure.*
*The concept of virtualization is physically implemented using the hypervisor modules, and the operation and processing of hypervisors are materialized by the OSS. In other words, hypervisor modules are installed on top of OSS, which act as an interface between hardware units and hypervisor packages."*

| Mallows-RMJ-PO-Top-2 | DPO |
|---|---|
| Here's a simplified version: | Here's a simplified version: |
| "Virtualization software, now called hypervisors, inherited many basic functions from early virtual machine operating systems. Hypervisors use hardware virtualization to provide customers with a dedicated computing, storage, and networking experience in the cloud, while sharing resources efficiently. | Virtualization technology, now known as a hypervisor, originated from early VM OS fundamentals. It enables cloud providers to efficiently share resources with customers by creating a simulated dedicated computing, storage, and networking experience. Hypervisor modules, built on Open-Source Software (OSS), act as an interface between hardware and hypervisor packages, physically implementing virtualization concepts. |
| The concept of virtualization is implemented through hypervisor modules, which are built on top of Operating System Software (OSS). OSS acts as an interface between hardware and hypervisor packages, enabling the operation and processing of hypervisors." | |
| I removed some redundant phrases, rephrased sentences for better clarity, and condensed the text to preserve the main ideas. Let me know if you'd like any further simplification! | |

**Notes:** These outputs are illustrative rather than fact-checked. Mallows-RMJ-PO-Top-2 exhibits higher fidelity to the source text, whereas DPO introduces semantic distortion during simplification.

