# OpenReview forum: "Beyond Pairwise: Empowering LLM Alignment With (Ranked) Choice Modeling"
_ICLR.cc/2026/Conference — ICLR 2026 Poster_

### Official Review · Reviewer_Zc22 · 2025-10-20

**Soundness:** 3
**Presentation:** 3
**Contribution:** 3
**Rating:** 8
**Confidence:** 3

**Summary:**

The paper introduces RCPO, a framework that generalizes LLM alignment beyond pairwise preferences. It connects preference optimization to choice modeling via MLE, unifies DPO and S-DPO (as MNL-PO-Discrete) under this framework, and derived loss functions for MNL and Mallows-RMJ models choice models towards single-best and top-k rankings.

The authors demonstrate its effectiveness by fine-tuning Llama-3-8B-Instruct and Gemma-2-9B-it on preference data derived from UltraFeedback and Skywork-Reward-V2-Llama-3.1-8B reward mode. The results consistently outperformed baselines like DPO/KPO and S-DPO.

The paper also covers practical tricks to implement the Mallows-RMJ objectives, like sigmoid-smoothing approximation for non-differentiable step functions, and use entropy to estimate the dispersion parameter.

**Strengths:**

The paper presents a principled and unified view connecting preference optimization, choice modeling, and MLE, bridging the literature on choice models with LLM alignment. The derivation of the new methods is very principled, and the presentation is clear.

The paper proposes two new methods (MNL and Mallows-RMJ based) for leveraging single-best and top-k rankings in preference optimization. The proposed methods are practical, providing practitioners with useful tools for exploring different sampling and ranking strategies in LLM post-training.

The experimental design and outcomes are overall convincing.

**Weaknesses:**

While the authors provide reasonable baselines for the pairwise and single-best scenarios, the evaluation of the top-k methods could be more robust by including something like a vanilla DPO method trained on all pairs implied by the top-2 ranking, especially given the impact of negative sampling on DPO; it is unclear how much of improvement from the “-Top2” methods come from a sheer negative sampling change.

Additionally, while the paper positions RCPO as a generalized framework and provides good insights on how choice modeling and MLE connect, the core methods derived feels a bit fragmented as their derivation requires several very specific assumptions. The paper feels more like a collection of methods derived under a similar principle. E.g., Citing  “For the baseline method, I used RCPO with MNL choice model for top-k choice” sounds a bit awkward.

**Questions:**

Could the authors provide further reasoning/evidence on the suitability of Skywork-Reward-V2 for labeling the complex reasoning tasks in AlpacaEval 2 and Arena-Hard? Asking because the qualitative example in Table 15 shows the aligned models replicating a mathematical error (x**2 − x − 8 = (x − 2)(x + 4)), and curious if that could be related to RM quality (i.e. did the preference labeler catch the error?)

---

> ### Author Response · Authors · 2025-11-25
> **Author Response**
>
> We thank you very much for your helpful comments. Please see our responses below.
>
> > **W1: Comparison against a vanilla DPO method trained on all pairs implied by the top-2 ranking.**
> >
>
> Thank you for mentioning this point. In fact, OpenAI reportedly collected rankings of K responses (ranging from 4 to 9) per prompt and trained the model on all $\binom{K}{2}$ pairs, at least during its development stage (see Ouyang et al. 2022, Appendix A.3). In fact, this mismatch between actual data format and the model input was in part what motivated our research.
>
> During the rebuttal period, we implemented this idea as DPO-AllPairs.  We find it shows weak performance in terms of both effectiveness and time efficiency.
>
> **Effectiveness.** As shown in Table Zc22-1 below, the DPO-AllPairs method performs not only substantially worse than our MNL-PO-Top2 and Mallows-RMJ-PO-Top2 methods, but also worse than the DPO method trained on sampled pairs.
>
> **Table Zc22-1: Evaluation Results for Llama-3-8B-Instruct**
>
> | Method | AlpacaEval 2 LC (%) | AlpacaEval 2 WR (%) | Arena-Hard WR (%) |
> |------------|-------------------------|--------------------------|-----------------------|
> | DPO | 41.24 (0.35) | 40.24 (1.66) | 32.6 (30.6, 34.7) |
> | DPO-AllPairs | 33.02 (0.24) | 38.47 (1.65) | 29.6 (27.3, 31.5) |
> | MNL-PO-Top2 | 40.12 (0.22) | 47.69 (1.67) | 35.8 (33.2, 38.2) |
> | Mallows-RMJ-PO-Top2 | 41.95 (0.26) | 53.01 (1.68) | 37.2 (35.0, 39.4) |
>
> **Time Efficiency.** DPO-AllPairs requires significantly more training time than our ranked choice methods. In our experiments, each top-2 ranking implies 7 pairwise comparisons. As a result, the training time of DPO-AllPairs grows around 7 times compared to DPO. In contrast, the training time for our ranked choice approach scales much more favorably, growing only approximately linearly with the assortment size $|S|$. Specifically, our $|S|=5$ setting required a training time increase of only about 2.5 times relative to DPO.
>
> **Table Zc22-2: Training Time on Llama-3-8B-Instruct**
>
> | Method | Training Time (Hours) |
> |------------|-------------------------|
> | DPO | 1.5 |
> | DPO-AllPairs  | 11.5 |
> | MNL-PO-Top2 | 3.5 |
> | Mallows-RMJ-PO-Top2 | 3.5 |
>
>
> We have also added relevant discussion of this method in Table 2 and Table 16 in the updated paper.
>
> Reference:
> * Ouyang, L., Wu, J., Jiang, X., Almeida, D., Wainwright, C., Mishkin, P., ... & Lowe, R. (2022). Training language models to follow instructions with human feedback. _Advances in neural information processing systems_, _35_, 27730-27744.
>
>
> > **W2: The core methods derived feels a bit fragmented as their derivation requires several very specific assumptions. The paper feels more like a collection of methods derived under a similar principle.**
>
> Thank you for the thoughtful comment. RCPO is proposed as a unified likelihood-based framework for a broad family of ranked choice models. In this sense, the main contribution of this paper is conceptual. The MNL-based and Mallows-RMJ variants are included as two illustrative examples, each representing a different way to represent the underlying preference parameters. Both reduce to the same optimization principle and share identical training and evaluation pipelines within the RCPO framework.
>
> We will revise the paper to make this unifying structure clearer—explicitly showing how each example fits into the same objective and satisfies the two framework conditions (reward sufficiency and MLE estimability).

---

> > ### Author Response · Authors · 2025-11-25
> > **Author Response**
> >
> > > **Q: Could the authors provide further reasoning/evidence on the suitability of Skywork-Reward-V2 for labeling the complex reasoning tasks in AlpacaEval 2 and Arena-Hard? Asking because the qualitative example in Table 15 shows the aligned models replicating a mathematical error $(x^2 - x - 8 = (x - 2)(x + 4))$, and curious if that could be related to RM quality (i.e. did the preference labeler catch the error?)**
> >
> > We wish to clarify that Skywork-Reward-V2 is **not** used as the evaluation judge. In these benchmarks, the judgments are produced by GPT-4.1-mini, not by Skywork-Reward-V2. Instead, Skywork-Reward-V2 was only used in our training data pipeline to rank multiple model responses. There are also established evaluation protocols for AlpacaEval 2 and Arena-Hard, as commonly practiced in recent literature (e.g., Meng et al., 2024, Chen et al., 2025, Zhao et al., 2025, Chen et al., 2025). Our evaluation methodology is based on the same protocol.
> >
> >
> > That said, your comment helped us realize that LLM-as-a-judge may be vulnerable to position bias. While AlpacaEval 2 applies random-swap strategy to mitigate this, it still exists on the instance-level, which is what led us to the example. More specifically, our example was picked by choosing from where the Mallows-RMJ-generated responses receiving favorable scores compared to the reference model (GPT-4-Turbo), and the reference-model-generated responses receiving favorable scores compared to DPO-generated ones, with the default order.  Inspired by your comment, we did the following additional checks:
> >
> > (i) We swapped the positions and found that the `equation` example no longer satisfies the criterion above. This means that while the GPT-4.1-mini judge did recogonize the mathematical error, it was overwhelmed by the positional bias in the example.
> >
> > (ii) We independently conducted an extra round of robustness checks and re-run the evaluation procedures with manually swapped positions of **all** responses in AlpacaEval 2. Our results are presented in the table below, which is largely the same as the one presented in the paper. They indicates that although GPT-4.1-mini is vulnerable to positional biases individually, the current evaluation protocols (with built-in random swaps) are still reliable at the aggregate level.
> >
> > (iii) We updated the example selection rule after explicitly taking into account the individual positional biase. We verify that while the equation example would no longer be selected, while the other three examples in the previous Table 12,13,14 remain valid. We therefore replace this example with others that suit the new criterion in the updated paper.
> >
> > | Method | AlpacaEval 2 LC (%) | AlpacaEval 2 WR (%) | AlpacaEval 2 LC (%) swap | AlpacaEval 2 WR (%) swap|
> > |------------|-------------------------|--------------------------|-----------------------|--------------------------|
> > | Mallows-RMJ-PO-Pairwise | 39.33 (0.28) | 48.71 (1.67)| 38.61 (0.29) | 49.48 (1.68) |
> > | MNL-PO-Discrete | 41.33 (0.29)| 48.08 (1.68) | 39.16 (0.34)| 47.23 (1.68) |
> > | Mallows-RMJ-PO-Discrete | 39.19 (0.28) | 51.17 (1.67)  | 37.72 (0.30) | 50.08 (1.69) |
> > | MNL-PO-Top2 | 40.12 (0.22) | 47.69 (1.67) | 39.51 (0.31) | 48.76 (1.68) |
> > | Mallows-RMJ-PO-Top2 | 41.95 (0.26) | 53.01 (1.68)  | 39.84 (0.30) | 52.17 (1.69)|
> >
> >
> >
> >
> > Reference:
> > * Meng, Y., Xia, M., & Chen, D. (2024). Simpo: Simple preference optimization with a reference-free reward. _Advances in Neural Information Processing Systems_, _37_, 124198-124235.
> > * Chen, H., Zhao, H., Lam, H., Yao, D. D., & Tang, W. (2025, January). MallowsPO: Fine-Tune Your LLM with Preference Dispersions. In _ICLR_.
> > * Zhao, H., Winata, G. I., Das, A., Zhang, S. X., Yao, D. D., Tang, W., & Sahu, S. (2024). Rainbowpo: A unified framework for combining improvements in preference optimization. _arXiv preprint arXiv:2410.04203_.
> > * Chen, P., Chen, X., Yin, W., & Lin, T. (2025). ComPO: Preference alignment via comparison oracles. _arXiv preprint arXiv:2505.05465_.

---

> > > ### Comment · Reviewer_Zc22 · 2025-11-27
> > >
> > > Thank you for your thoughtful response, and I appreciate the additional experiment on the vanilla method, as well as the analysis on position bias.
> > >
> > > As reflected in my original scores, I think this is a good paper and I would like to keep my current score.

---

### Official Review · Reviewer_XFCx · 2025-10-30

**Soundness:** 3
**Presentation:** 3
**Contribution:** 3
**Rating:** 8
**Confidence:** 2

**Summary:**

This paper introduces Ranked Choice Preference Optimization (RCPO), a unified framework designed to improve Large Language Model (LLM) alignment by leveraging richer human feedback, such as top-k rankings, instead of relying solely on pairwise comparisons.
Empirical evaluations on Llama-3-8B and Gemma-2-9B show that RCPO variants consistently outperform competitive baselines on major benchmarks, with the rank-based Mallows-RMJ model achieving particularly strong results using top-2 feedback.

**Strengths:**

The paper presents the generalized framework: Ranked Choice Preference Optimization (RCPO) which subsumes several existing pairwise methods, including Direct Preference Optimization (DPO), SimPO, and R-DPO, viewing them as special cases.

It is not limited to pairwise comparisons, but can directly leverage richer human feedback formats, such as multi wise comparisons, single-best selections from a set, and top-k rankings.

Empirical studies on Llama-3-8B-Instruct and Gemma-2-9B-it show that RCPO variants consistently outperform competitive baselines (including DPO, SimPO, and IPO) on widely adopted benchmarks like AlpacaEval 2 and Arena-Hard. The best-performing variant, Mallows-RMJ-PO-Top2, surpassed the strongest non-RCPO baseline (IPO) by significant margins.

The improvements are shown on Llama-3 and Gemma-2 base models.

**Weaknesses:**

The paper only shows experimental results with Top-2 feedback. The authors say that they stop at Top-2 since the performance does not always increase with more feedback. It would be very interesting to see when this happens. Only seeing Top-2 results, when the obvious next step would be to try a higher number Top-k, gives me that the authors are trying to hide something.

The experiments are conducted only on relatively smaller models, specifically Llama-3-8B-Instruct and Gemma-2-9B-it. The paper does not demonstrate whether the gains observed with RCPO scale to larger models.

**Questions:**

Some of the equations in the appendix move out of bounds. You should fix that.

Did you run any Top-k experiments with k > 2? If not, why not? Just that it might get worse doesn’t sound like a good reason. It might get better as well.

---

> ### Author Response · Authors · 2025-11-25
> **Author Response**
>
> We thank you very much for your helpful comments. Please see our responses below.
>
> > **W1 and Q2: Larger $k$.**
> > The paper only shows experimental results with Top-2 feedback. The authors say that they stop at Top-2 since the performance does not always increase with more feedback. It would be very interesting to see when this happens. Only seeing Top-2 results, when the obvious next step would be to try a higher number Top-k, gives me that the authors are trying to hide something.
> > Did you run any Top-k experiments with k > 2? If not, why not? Just that it might get worse doesn’t sound like a good reason. It might get better as well.
>
> Thank you for the thoughtful suggestion. We did not conduct experiments beyond $k=2$ because of time and computation resource limitation and simply put our hypotheses briefly in the paper. However, we have followed your suggestion and conducted new experiments for larger $k$. Let us next report the findings from two aspects: effectiveness and time efficiency. The results are contained in Table XFCx-1.
>
> **Effectiveness.** We find that, as conjectured, a large $k$ may not necessarily improve performance. For example, when $|S|=5$, $k=2$ is the best. One possible reason is that higher values of $k$ make the alignment process more reliant on high-quality data. Specifically, when $k$ is small, selecting the top-$k$ items is relatively easy, but as $k$ becomes larger, distinguishing the relative order among the remaining candidates may become more difficult. Hence a too large $k$ can introduce additional noise or place unnecessary burden on annotators. This finding is consistent with the prior literature in a best-item selection context (Feng and Tang, 2023). It also highlights the benefit of top-k feedback compared to the more extreme options, such as pairwise comparisons or full-ranking feedback.
>
>
> **Table XFCx-1: Evaluation Results of Varied $k$ with $|S|=5$**
> | Method | AlpacaEval 2 LC (%) | AlpacaEval 2 WR (%) | Arena-Hard WR (%) |
> |------------|-------------------------|--------------------------|------------------------|
> | Mallows-RMJ-PO-Discrete | 39.19 (0.28) | 51.17 (1.67) | 36.3 (34.6, 37.9) |
> | Mallows-RMJ-PO-Top2 | **41.95 (0.26)** | **53.01 (1.68)** | 37.2 (35.0, 39.4) |
> | Mallows-RMJ-PO-Top3 | 41.05 (0.25) | 52.59 (1.69) | **37.7 (35.8, 39.8)** |
> | Mallows-RMJ-PO-Top4 | 39.92 (0.26) | 51.08 (1.68) | 37.4 (35.5, 39.2) |
>
>
> **Time efficiency.** As shown in Table XFCx-2 below, we find the training time is insensitive to the value of $k$.
>
> **Table XFCx-2: Evaluation Results of Varied $k$**
> | Method | Training Time (Hours) |
> |------------|-------------------------|
> | Mallows-RMJ-PO-Discrete | 3.5 |
> | Mallows-RMJ-PO-Top2 | 3.5 |
> | Mallows-RMJ-PO-Top3 | 3.5  |
> | Mallows-RMJ-PO-Top4 | 3.5  |
>
>
> We have also added these in Table 4 in the updated paper.
>
> Reference:
> * Feng, Y., & Tang, Y. (2025). A mallows-type model for preference learning from (ranked) choices.  _Available at SSRN 4539900_.
>
> > **W2: Larger models.**
> > The experiments are conducted only on relatively smaller models, specifically Llama-3-8B-Instruct and Gemma-2-9B-it. The paper does not demonstrate whether the gains observed with RCPO scale to larger models.
>
> Thank you for the suggestion. We evaluate on Llama-3-8B-Instruct and Gemma-2-9B-it, which align with common practice for preference optimization (e.g., Yuan et al., 2023; Rafailov et al., 2023; Hone et al., 2024; Park et al., 2024; Meng et al., 2024). Full 30B+ training is outside our current compute budget, but would definitely be interesting direction for future research.
>
>
>
> Reference:
> * Yuan, H., Yuan, Z., Tan, C., Wang, W., Huang, S., & Huang, F. (2023). Rrhf: Rank responses to align language models with human feedback. _Advances in Neural Information Processing Systems_, _36_, 10935-10950.
> * Rafailov, R., Sharma, A., Mitchell, E., Manning, C. D., Ermon, S., & Finn, C. (2023). Direct preference optimization: Your language model is secretly a reward model. _Advances in neural information processing systems_, _36_, 53728-53741.
> * Hong, J., Lee, N., & Thorne, J. (2024). Orpo: Monolithic preference optimization without reference model. _arXiv preprint arXiv:2403.07691_.
> * Park, R., Rafailov, R., Ermon, S., & Finn, C. (2024). Disentangling length from quality in direct preference optimization. _arXiv preprint arXiv:2403.19159_.
> * Meng, Y., Xia, M., & Chen, D. (2024). Simpo: Simple preference optimization with a reference-free reward. _Advances in Neural Information Processing Systems_, _37_, 124198-124235.
>
>
> > **Q1: Some of the equations in the appendix move out of bounds.**
>
> Thank you for the good catch. We have fixed these issues in the updated paper.

---

### Official Review · Reviewer_fgCa · 2025-11-01

**Soundness:** 2
**Presentation:** 2
**Contribution:** 2
**Rating:** 2
**Confidence:** 4

**Summary:**

Pairwise comparison forms the foundation of model alignment, and despite the emergence of alternative approaches such as KTO and joint preference optimization, pairwise methods remain dominant in the community. In this work, the authors propose a framework that extends beyond traditional pairwise comparisons by incorporating rank-order information. They introduce two complementary frameworks under this paradigm (called RCPO) and evaluate them both against each other and against established baselines such as DPO, demonstrating consistent improvements from the proposed approach.

**Strengths:**

I appreciate an alternate modeling for the preference alignment, the associated literature, and the corresponding MLE estimator formulations in the LLM alignment setup, with a given optimal reward structure. I also appreciate the gradient analysis, and to my understanding, the MNL style approach seems close to one-vs-many, where the MALLOWS-RMJ MODEL seems close to one-vs-one, but weighted by the size of the universe.

**Weaknesses:**

### Overall Comments
This work would benefit from more experiments and broader comparisons with related methods. Given the current limitations, I don’t think it’s yet ready for this venue.

---

### Related Work on Going Beyond Pairwise Ranking
The paper should include quantitative comparisons with several recent methods that also move beyond pairwise preference modeling:

- **LiPO-λ (Listwise Preference Optimization through Learning-to-Rank):** Uses LambdaLoss with permutation-aware weighting and improves over DPO on Reddit TL;DR and Anthropic-HH.
- **KPO (K-order Ranking Preference Optimization):** Extends Plackett–Luce models with adaptive \(K\) selection and shows better sample efficiency on recommendation benchmarks.
- **PPA (Permutative Preference Alignment):** Optimizes differentiable NDCG-based objectives and outperforms pairwise methods on AlpacaEval.
- **AMP/MDPO (Automated Multi-level Preference Optimization):** Handles multi-level preferences and achieves strong results on hallucination reduction benchmarks.

Adding comparisons with these works would make the evaluation more complete.

---

### More Models
The results are currently based on only two models. It would help to include additional models such as **Qwen2.5** or **Mistral** to test how well the method generalizes.

---

### More Datasets
It would also strengthen the paper to include more datasets, such as **TL;DR**, **Anthropic-HH**, **UltraFeedback**, and **WildBench** (which is more challenging than AlpacaEval 2.0).

---

### Limited and Inconsistent Improvements
The improvements are not consistent across models or datasets.
For example:
- On **LLaMA**, LC–SimPO performs much better.
- On **Gemma**, **DPO** performs better.
- On **Arena-Hard**, the improvements are small and may not be significant.

---

### Ablation Studies
1. The paper mentions that \(k=2\) performs best, but it would be useful to show how performance changes as \(|S|\) and \(k\) vary together.
2. It would help to test what happens if the \(|S|-i\) weighting in the Mallows RMJ model is removed, since that still represents a valid one-vs-one setup.
3. The paper should also study **sample efficiency**—how performance changes as the amount of training data increases (e.g., WR and LC at different data budgets).
4. Sensitivity analysis on $\beta$

**Questions:**

- It is important to consider the practical challenges of collecting ranked-choice data, both from an implementation standpoint and a psychological one. Given the current data collection pipelines used by major industry LLMs, adopting a ranking-based approach would represent a significant paradigm shift. Moreover, the usefulness of rank information—specifically, the choice of k, |S|, is highly task-dependent. This raises questions about whether the additional computational and monetary costs of obtaining such data are justified, especially when the observed performance gains appear relatively modest.

---

> ### Author Response · Authors · 2025-11-25
> **Author Response**
>
> Thank you for the constructive comments. We have expanded the experiments and robustness checks accordingly. Our detailed responses are below.
>
> > **W1: Related Work on Going Beyond Pairwise Ranking.**
>
>
> Thank you for pointing us to these references. We have added citations and expanded the discussion in the revised paper.
>
> * LiPO-$\lambda$ and PPA: These approaches rely on scalar-valued labels (scores) for each response. In contrast, DPO and our RCPO framework work with ordinal feedback such as comparisons, top-k lists, or rankings. Since the underlying feedback structures differ, direct apple-to-apple comparison does not apply here.
>
>
> * KPO: Thank you for bringing this related method to our attention. We carefully examined the method and found that it leads to the same training objective as MNL-PO-Topk when applied to our setting, although it was originally proposed for a different application (i.e., ranking items in recommendation systems). Plus, KPO is derived from the Plackett-Luce model, while MNL-PO-Topk follows from our unifying RCPO framework under random utility with Gumbel noise. When applied to LLM alignment, they behave equivalently.
>
>
> * AMP/MDPO: These methods convert each ranking into all pairwise comparisons and then apply a DPO-style loss. We implemented this idea and added a new method referred to as DPO-AllPairs in our new numerical studies. As shown in Table fgCa-1 and Table fgCa-2 below, it requires much longer training time yet produces weaker alignment. This highlights the benefit of training **directly** on top-k or full ranking feedback instead of reducing rankings to synthetic pairs.
>
> **Table fgCa-1: Evaluation Results for Llama-3-8B-Instruct**
>
> | Method | AlpacaEval 2 LC (%) | AlpacaEval 2 WR (%) | Arena-Hard WR (%) |
> |------------|-------------------------|--------------------------|-----------------------|
> | DPO | 41.24 (0.35) | 40.24 (1.66) | 32.6 (30.6, 34.7) |
> | DPO-AllPairs (MDPO) | 33.02 (0.24) | 38.47 (1.65) | 29.6 (27.3, 31.5) |
> | MNL-PO-Top2 | 40.12 (0.22) | 47.69 (1.67) | 35.8 (33.2, 38.2) |
> | Mallows-RMJ-PO-Top2 | 41.95 (0.26) | 53.01 (1.68) | 37.2 (35.0, 39.4) |
>
> **Table fgCa-2: Training Time on Llama-3-8B-Instruct**
>
> | Method | Training Time (Hours) |
> |------------|-------------------------|
> | DPO | 1.5 |
> | DPO-AllPairs (MDPO) | 11.5 |
> | MNL-PO-Top2 | 3.5 |
> | Mallows-RMJ-PO-Top2 | 3.5 |
>
>
> > **W2: More Models**
>
> We added Mistral-7B-Instruct as a third base model. The results in Table fgCa-3 are consistent with our findings for Llama-3-8B and show that RCPO methods continue to offer strong improvements. We have also added these into the Table 3 in the updated paper.
>
> **Table fgCa-3: Evaluation Results for Mistral-7B-Instruct**
>
> | Method | AlpacaEval 2 LC (%) | AlpacaEval 2 WR (%) | Arena-Hard WR (%) |
> |------------|-------------------------|--------------------------|------------------------|
> | Base Model | 14.53 (0.44) | 11.94 (1.09) | 10.8 (9.5, 12.0) |
> | SimPO | 26.32 (0.38) | 30.13 (1.54) | **19.3** (17.8, 20.6) |
> | DPO | 23.32 (0.39) | 19.91 (1.34) | 16.8 (15.5, 18.3) |
> | Mallows-RMJ-PO-Top2 | **29.57** (0.22) | **37.58** (1.68) | 16.9 (15.4, 18.3) |
>
> > **W3: More Datasets**
>
> Our previous evaluations on AlpacaEval 2 and Arena-Hard are both **out-of-distribution test**. We have included the Ultrafeedback test dataset as an **in-distribution test**. Again, we evaluate performance by measuring the win rate of generated responses against the chosen responses in the dataset, using GPT-4.1-mini as the judge. As shown in Table fgCa-4 below, our RCPO methods all perform well and the Mallows-RMJ-PO-Top2 is the best, which is consistent with our earlier results. We have added these into the Table 2 in the updated paper.
>
> **Table fgCa-4: Evaluation Results on Ultrafeedback Test Data**
>
> | | Base Model| CPO | IPO | ORPO | RRHF | SLiC-HF | KTO | DPO | R-DPO | SimPO | DPO-AllPairs | Mallows-RMJ-PO-Pairwise | MNL-PO-Discrete | Mallows-RMJ-PO-Discrete | MNL-PO-Top2 | Mallows-RMJ-PO-Top2 |
> |-----------------|------|-----|-----|------|------|---------|-----|-----|-------|-------|---------------|-------------------|--------------|------------------|-----------|--------------|
> | Ultrafeedback WR (%) | 42.51 (1.04) | 58.71 (1.05) | 58.44 (1.05) | 51.68 (1.06) | 44.28 (1.06) | 44.09 (1.06) | 53.33 (1.06) | 62.36 (1.03) | 60.68 (1.05) | 50.17 (1.09) | 51.95 (1.08) | 66.28 (1.02) | 64.64 (1.03) | 67.56 (1.01) | 64.01 (1.04) | **68.91** (1.00) |

---

> > ### Author Response · Authors · 2025-11-25
> > **Author Response**
> >
> > > **W4:  Limited and Inconsistent Improvements**
> >
> > The main contribution of this work is not a single algorithm, but rather a general framework that connects LLM alignment with ranked choice modeling. It unifies a few existing preference optimization methods, but more importantly, generates new preference optimization methods in a principled way. The specific examples (e.g., Mallows-RMJ) mainly serve as illustrative examples rather than endpoints.
> >
> > As shown in Table 2, methods derived from our framework consistently outperform those that lie outside it. We view this as strong evidence of the framework’s promise, which is that the right feedback structure paired with the right choice model yields more effective alignment.
> >
> > > **W5:  Ablation Studies**
> > >
> >
> > We have added more ablation studies as requested.
> >
> > > 1. Varied $k$ and $|S|$.
> >
> > We have conducted comprehensive numerical studies on varied $k$ and $|S|$. Next, we will show the findings from two aspects: effectivenss (response quality) and time efficiency.
> >
> > **Effectiveness.** As shown in Table fgCa-5 below, we have the following general findings:
> >
> > * Increasing $|S|$ from two to three and five (current setting) generally leads to better performance. Notably, even $|S|=3$ can achieve significant improvement over pairwise ($|S|=2$). One possible reason is that incorporating more negative samples enables the language model to learn better distinctions.
> > * A large $k$ may not necessarily improve performance. For example, when $|S|=5$, $k=2$ is the best. One possible reason is that higher values of $k$ make the alignment process more reliant on high-quality data. Specifically, when $k$ is small, selecting the top-$k$ items is relatively easy, but as $k$ becomes larger, distinguishing the relative order among the remaining candidates may become more difficult. Hence a too large $k$ can introduce additional noise or place unnecessary burden on annotators. This finding is conceptually consistent with the prior literature in a best-item selection context (Feng and Tang 2023). It also highlights the benefit of top-k feedback compared to the more extreme options, such as pairwise comparisons or full-ranking feedback.
> >
> > In general, we believe that a moderately small $(|S|,k)$ will stand in the sweet spot between keeping feedback simple and making use of data efficiently, as discussed in the paper.
> >
> >
> >
> > **Table fgCa-5: Evaluation Results of Varied $k$ and $|S|$**
> > | Method | AlpacaEval 2 LC (%) | AlpacaEval 2 WR (%) | Arena-Hard WR (%) |
> > |------------|-------------------------|--------------------------|------------------------|
> > | DPO           | 41.24 (0.35) |40.24 (1.66)  | 32.6 (30.6,34.7)  |
> > | Mallows-RMJ-PO-Pairwise | 39.33 (0.28) | 48.71 (1.67) | 36.5 (34.3,38.6)  |
> > | assort3-Mallows-RMJ-PO-Discrete | 40.28 (0.27) | 50.49 (1.68) | 36.3 (34.1, 38.8) |
> > | assort3-Mallows-RMJ-PO-Top2 | 39.57 (0.27) | 49.65 (1.68) | 36.9 (34.7, 39.0) |
> > | assort5-Mallows-RMJ-PO-Discrete | 39.19 (0.28) | 51.17 (1.67) | 36.3 (34.6, 37.9) |
> > | assort5-Mallows-RMJ-PO-Top2 | **41.95 (0.26)** | **53.01 (1.68)** | 37.2 (35.0, 39.4) |
> > | assort5-Mallows-RMJ-PO-Top3 | 41.05 (0.25) | 52.59 (1.69) | **37.7 (35.8, 39.8)** |
> > | assort5-Mallows-RMJ-PO-Top4 | 39.92 (0.26) | 51.08 (1.68) | 37.4 (35.5, 39.2) |
> >
> > **Time efficiency.** As shown in Table fgCa-6 below, we have the following general findings:
> > * The training time is not sensitive to the underlying choice model (e.g., MNL vs. Mallows-RMJ).
> > * The training time is also not very sensitive to the value of $k$.
> > * The training time grows approximately linearly with the assortment size $|S|$. The reason is that when $|S|$ is larger, less responses can be contained in one batch, and therefore more time to complete every epoch.
> >
> > **Table fgCa-6: Evaluation Results of Varied $k$ and $|S|$**
> > | Method | Training Time (Hours) |
> > |------------|-------------------------|
> > | DPO           | 1.5 |
> > | Mallows-RMJ-PO-Pairwise | 1.5 |
> > | assort3-Mallows-RMJ-PO-Discrete | 2.3 |
> > | assort3-Mallows-RMJ-PO-Top2 | 2.3 |
> > | assort5-Mallows-RMJ-PO-Discrete | 3.5 |
> > | assort5-Mallows-RMJ-PO-Top2 | 3.5 |
> > | assort5-Mallows-RMJ-PO-Top3 | 3.5  |
> > | assort5-Mallows-RMJ-PO-Top4 | 3.5  |
> >
> >
> > We have included these new results at Table 4 in the updated paper.

---

> > > ### Author Response · Authors · 2025-11-25
> > > **Author Response**
> > >
> > > > 2. Removing $(|S|-i)$ weight.
> > >
> > > We want to clarify that the **Mallows-RMJ model is not a one-vs-one approach**. As shown in the paper, $d(\mu^k, S) = \sum_{i=1}^{k-1} \mathbb{I} \lbrace \mu_0^{-1}(y_i \mid x) > \mu_0^{-1}(y_{i+1} \mid x) \rbrace (|S| - i) + \sum_{y_j \in S \setminus \lbrace y_1, \dots, y_k \rbrace} \mathbb{I} \lbrace \mu_0^{-1}(y_k \mid x) > \mu_0^{-1}(y_j \mid x) \rbrace $
> > > , for items ranked in the top-$k$, each item $y_i$ is only compared with its immediate successor $y_{i+1}$. However, the last top-$k$ item, $y_{k}$, is compared against all items not in the top-$k$. This demonstrates that the model is not a one-vs-one setup.
> > >
> > > In addition, those weights are important components, since if the weights are different, one can no longer aggregate into the simple ranked choice probabilities using the same principle.
> > >
> > > > 3. Sample efficiency.
> > >
> > > We have evaluated the performance as the amount of training data increases. As shown in Table fgCa-7, Mallows-RMJ-PO-Top2 can achieve compariable performance when the data size is much smaller. This shows our method can achieve sample efficiency. We have included this new result at Figure 3 in the updated paper.
> > >
> > > **Table fgCa-7: Sample Efficiency of Mallows-RMJ-PO-Top2**
> > >
> > > | Training Steps | AlpacaEval 2 LC (%) | AlpacaEval 2 WR (%) |
> > > |------------|-------------------------|--------------------------|
> > > | 0 | 24.76 (0.42) | 24.40 (1.44) |
> > > | 100 | 30.01 (0.32) | 35.22 (1.60) |
> > > | 200 | 36.53 (0.28) | 50.64 (1.66) |
> > > | 300 | 38.20 (0.32) | 50.05 (1.68) |
> > > | 400 | 40.28 (0.24) | 51.61 (1.67) |
> > > | 529 | 41.95 (0.26) | 53.01 (1.68) |
> > > | DPO | 41.24 (0.35) | 40.24 (1.66) |
> > >
> > >
> > > > 4. Sensitivity analysis on $\beta$.
> > >
> > > We have conducted sensitivity analysis on $\beta$. The results are shown in Table fgCa-8 below. We have also added these in Figure 4 in the updated paper.
> > >
> > > **Table fgCa-8: Sensitivity analysis on $\beta$ of Mallows-RMJ-PO-Top2**
> > >
> > > | Metric | β = 0.003 | β = 0.01 | β = 0.03 | β = 0.05 |
> > > |--------|-----------|----------|----------|----------|
> > > | AlpacaEval 2 LC (%) |  41.88 | **41.95** | 39.71 | 41.18 |
> > > | AlpacaEval 2 WR (%) | 52.79 | **53.01** | 51.03 | 51.21 |
> > >
> > > > **Q**
> > > > It is important to consider the practical challenges of collecting ranked-choice data, both from an implementation standpoint and a psychological one.
> > >
> > > * Implementation: as shown above, $k$ does not affect training time but can meaningfully improve the alignment effectivenss. Furthermore, ranked choice data is more sample efficient. As shown in Table fgCa-7 above, using top-2 choice data can reduce the sample size by around 40% compared with pairwise training.
> > > * Psychological standpoint: asking human annotators to rank many responses (larger $k$) can impose a higher cognitive burden. However, Table fgCa-5 shows a small $k$ (e.g., $k=2$) may be already enough to signficantly improve the effectiveness while keeping the task simple.
> > >
> > >
> > > > Given the current data collection pipelines used by major industry LLMs, adopting a ranking-based approach would represent a significant paradigm shift.
> > >
> > > First, we wish to clarify that some industry practice does collect ranking data in its raw form. For example, OpenAI reportedly collected rankings of K responses (ranging from 4 to 9) per prompt and trained the model on all $\binom{K}{2}$ pairs, at least during its development stage (see Ouyang et al. 2022, Appendix A.3). In fact, this mismatch between actual data format and the model input was in part what motivated our research.
> > >
> > > More broadly, while pairwise comparison remains common, our results show that directly using richer ranking feedback can provide clear benefits. We hope that may encourage more use of structured preference data in the LLM alignment.

---

> > > > ### Author Response · Authors · 2025-11-25
> > > > **Author Response**
> > > >
> > > > > Moreover, the usefulness of rank information—specifically, the choice of k, |S|, is highly task-dependent. This raises questions about whether the additional computational and monetary costs of obtaining such data are justified, especially when the observed performance gains appear relatively modest.
> > > >
> > > > Let us respond to the cost problem from two aspects: cost of data collection and cost of model training.
> > > >
> > > > * Cost of data collection: As shown in Table fgCa-7, training on ranked-chocie data can be much more sample efficient than training on pairwise data. As a result, the monetary costs of data collection may even decrease in practice.
> > > >
> > > > * Cost of computation: Based on our experience, the cost of computation largely depends on the GPU time in training. As shown in Table fgCa-6 above, the training time is insensitive to the value of $k$ or the choice model itself. While training time grows roughly linearly in the assortment size, we find that even a small value of $|S|=3$ delivers clear performance gains relative to pairwise training while keeping computational demands manageable.
> > > >
> > > > In light of the studies above, we believe that in general, a moderately small $(|S|,k)$ can achieve an improved balance among (i) model performance; (ii) data/computational cost; and (iii) psychological burden on the feedback providers.
> > > >
> > > > Reference:
> > > > * Ouyang, L., Wu, J., Jiang, X., Almeida, D., Wainwright, C., Mishkin, P., ... & Lowe, R. (2022). Training language models to follow instructions with human feedback. _Advances in neural information processing systems_, _35_, 27730-27744.

---

### Comment · Area_Chair_k9AY · 2025-11-26

Dear Reviewers,

Thank you for sharing your valuable insights and expertise, which have played an important role in the review process.

In response to the initial feedback, the authors have submitted a detailed rebuttal addressing the comments raised by the reviewers.

I would appreciate it if you could carefully review their response and consider how it may affect your initial evaluation.

Please feel free to share your updated thoughts or any additional comments after reviewing the rebuttal.

Thank you again for your time and contributions.

---

### Author Response · Authors · 2025-12-01
**Author Summary**

We thank the reviewers for their constructive feedback and recognition of our work. The discussion has been very valuable and has strengthened the paper.

To summarize, we propose Ranked Choice Preference Optimization (RCPO), a unified framework that casts LLM alignment as maximum-likelihood training through ranked choice models, allowing models to learn directly from richer feedback such as single-best, top-k, and general multiwise rankings. It subsumes popular pairwise methods like DPO/SimPO, and we instantiate new methods using MNL and Mallows-RMJ models under single-best and top-k settings. Comprehensive experiments demonstrate their systematic improvements in alignment performance on multiple LLMs and datasets.

The reviewers' suggestions mainly centered on clarification, completeness, and robustness checks. Prompted by reviewers' suggestions, we conducted extensive new experiments and integrated them into the revised paper:

* We expanded the **robustness checks**, including a new base model (Mistral-7B-Instruct), a new evaluation dataset (Ultrafeedback test data), and a new baseline (DPO-AllPairs).
* We conducted more comprehensive ablation studies that provide **deeper insights** into our framework. Specifically, we investigate:
    - the impact of ranked choice length and assortment size,
    - the sample efficiency of training on ranked choice data,
    - the effect of the β parameter, and
    - the impact of different treatments of ranked responses.
* We also revised the paper to **improve its clarity** and correct formatting issues.

We believe these updates address all reviewer concerns. During the rebuttal period, Reviewer Zc22 indicated satisfaction with our response.


We thank the AC and reviewers for their time, input, and consideration. We appreciate the opportunity to improve our work during the rebuttal period.

---

### Meta-Review · Area_Chair_neNq · 2026-01-06

**Summary:**

This work proposed Ranked Choice Preference Optimization (RCPO), a unified alignment framework that generalizes traditional pairwise preference optimization. From the viewpoint of maximum-likelihood training through ranked choice models, the model can learn directly from richer feedback such as single-best, top-k, and general multiwise rankings, which supports both utility-based and rank-based models. Empirical evaluations on Llama-3-8B and Gemma-2-9B show that RCPO consistently outperforms baselines on major benchmarks.

The main strength and novelty is that the framework for preference optimization is principled and unified, generalizes traditional pairwise preference optimization, and enables richer feedback. Also, the performance gain over baselines model are significant. The main weaknesses concerns were (1) the results are based on just two base models, where the authors provided results with Mistral-7B-Instruct are provided. However, the base models are not the most recent, and it would be better to include more recent models like Qwen-series models, which leaves a weakness of this paper. (2) More non-pairwise-preference works should be compared, and the rebuttal included more comparisons as required. (3) Ablation studies such as top-k incomplete are not comprehensive, and the authors further provided performance sensitivity results.

Most of the concerns are well addressed, while the experiments with base models are not very comprehensive and remain a weakness. Considering the contributions of the principled and unified framework and there is no strong weakness after rebuttal, the final recommendation is accept.

**Reviewer Concerns:**

The main weaknesses concerns were (1) the results are based on just two base models, where the authors provided results with Mistral-7B-Instruct are provided. However, the base models are not the most recent, and it would be better to include more recent models like Qwen-series models, which leaves a weakness of this paper. (2) More non-pairwise-preference works should be compared, and the rebuttal included more comparisons as required. (3) Ablation studies such as top-k incomplete are not comprehensive, and the authors further provided performance sensitivity results.

**Reviewer Scores:**

As most of the concerns were addressed while the base models are not the most recent, the reviewer who gave 2 may increase the score to 4.

---

### Decision · Program_Chairs · 2026-01-26

Accept (Poster)